**Seasonal variation of aerosol iron solubility in coarse and fine particles at an inland**
**city in northwestern China**
Huanhuan Zhang,[1,2] Rui Li,[1] Chengpeng Huang,[3] Xiaofei Li,[4] Shuwei Dong,[1] Fu Wang,[3]
Tingting Li,[1] Yizhu Chen,[1] Guohua Zhang,[1] Yan Ren,[3] Qingcai Chen,[4] Ru-jin Huang,[5] Siyu
Chen,[6] Tao Xue,[7] Xinming Wang,[1] Mingjin Tang[1,2,*]
[1] State Key Laboratory of Organic Geochemistry, Guangdong Key Laboratory of
Environmental Protection and Resources Utilization, and Guangdong-Hong Kong-Macao Joint
Laboratory for Environmental Pollution and Control, Guangzhou Institute of Geochemistry,
Chinese Academy of Sciences, Guangzhou, China
[2] College of Earth and Planetary Sciences, University of Chinese Academy of Sciences, Beijing,
China
[3] Longhua Center for Disease Control and Prevention of Shenzhen, Shenzhen, China
[4] School of Environmental Science and Engineering, Shaanxi University of Science and
Technology, Xi'an, China
[5] State Key Laboratory of Loess and Quaternary Geology, Institute of Earth Environment,
Chinese Academy of Sciences, Xi'an, China
[6] College of Atmospheric Sciences, Lanzhou University, Lanzhou, China
[7] School of Public Health, Peking University, Beijing, China
* Correspondence: Mingjin Tang (mingjintang@gig.ac.cn)

**Abstract**

This work investigated seasonal variation of aerosol iron (Fe) solubility for coarse (>1 μm) and fine (<1 μm) particles at Xi'an, a megacity in northwestern China impacted by anthropogenic emission and desert dust. Total Fe concentrations were lowest in summer and similar in other seasons for coarse particles, while lowest in summer and highest in spring for fine particles; for comparison, dissolved Fe concentrations were higher in autumn and winter than spring and summer for coarse particles, while highest in winter and lowest in spring and summer for fine particles. Desert dust aerosol was always the major source of total Fe for both coarse and fine particles in all the four seasons, but it may not be the dominant source for dissolved Fe. Fe solubility was lowest in spring for both coarse and fine particles, and highest in winter for coarse particles and in autumn for fine particles. In general aerosol Fe solubility was found to be higher in air masses originating from local and nearby regions than those arriving from desert regions after long-distance transport. Compared to coarse particles, Fe solubility was similar for fine particles in spring but significantly higher in the other three seasons, and at a given aerosol pH range Fe solubility was always higher in fine particles. Aerosol Fe solubility was well correlated with relative abundance of aerosol acidic species, implying aerosol Fe solubility enhancement by acid processing; moreover, such correlations were better for coarse particles than fine particles in all the four seasons. Fe solubility was found to increase with relative humidity and acid acidity for both coarse and fine particles at Xi'an, underscoring the importance of aerosol liquid water and aerosol acidity in regulating Fe solubility via chemical processing.

# 1 Introduction

Deposition of aerosol particles is a major external source of dissolved iron (Fe) in many open oceans (Boyd and Ellwood, 2010; Tagliabue et al., 2017), significantly affecting primary productions in these regions (Moore et al., 2009; Tang et al., 2021) and thus the global carbon cycle (Martin, 1990; Jickells et al., 2005). Dissolved Fe has also been recognized as an important source of reactive oxygen species in aerosol particles via mechanisms such as the Fenton reaction (Zhang et al., 2008; Fang et al., 2017; Wang et al., 2022) and thus may have adverse impacts on human health (Kelly, 2003; Abbaspour et al., 2014). In addition, dissolved Fe could catalyze aqueous oxidation of $SO_2$ (Martin and Good, 1991; Alexander et al., 2009; Huang et al., 2014), leading to the formation of sulfate, a major secondary species in aerosol particles. The various impacts of aerosol Fe are largely determined by its fractional solubility (often abbreviated as solubility) which is the ratio of dissolved Fe to total Fe.

Due to the impacts of dissolved aerosol Fe on ocean biogeochemistry and human health, a number of studies have been conducted in the last 2-3 decades (Chen and Siefert, 2004; Baker and Jickells, 2006; Kumar et al., 2010; Sholkovitz et al., 2012; Mahowald et al., 2018; Meskhidze et al., 2019; Zhu et al., 2020; Baker et al., 2021; Ito et al., 2021), leading to significant advances in our knowledge of aerosol Fe solubility and sources of aerosol dissolved Fe. For examples, many studies (Baker and Jickells, 2006; Sholkovitz et al., 2012) observed the inverse relationship between Fe solubility and total aerosol Fe. It has been recently realized that non-desert-dust sources, such as anthropogenic emissions and biomass burning, can be important for dissolved aerosol Fe in many regions (Sholkovitz et al., 2009; Ito et al., 2019;

Hamilton et al., 2020; Liu et al., 2022), though their contributions to total aerosol Fe are usually
minor. Furthermore, atmospheric aging processes, such as acid processing and organic
complexation, may substantially enhance solubility of Fe in desert dust and coal fly ash (Paris
et al., 2011; Shi et al., 2012; Chen and Grassian, 2013; Li et al., 2017).
Despite significant progress, it remains difficult for modelling studies to reproduce the
wide range of Fe solubility observed for ambient aerosols (Mahowald et al., 2018; Meskhidze
et al., 2019). The relative contribution of non-desert-dust sources, versus desert dust, to
dissolved aerosol Fe is still rather uncertain (Myriokefalitakis et al., 2018; Ito et al., 2019). In
addition, the impacts of chemical processing (especially organic complexation) on aerosol Fe
solubility is yet to be quantified for ambient aerosols. Further field measurements are needed
to reduce the uncertainties in aerosol Fe solubility, in order to better understand the impacts of
aerosol Fe on marine biogeochemistry and human health.
Sources, compositions and physicochemical properties are very different for coarse (>1
μm) and fine (<1 μm) particles (Seinfeld and Pandis, 2016). Therefore, aerosol Fe solubility
may differ significantly and is regulated by different sources or processes for coarse and fine
particles, as found by previous work (Sakata et al., 2022; Zhang et al., 2022). In addition, both
sources and chemical processes of aerosol particles exhibit significant variability for different
seasons, consequently leading to seasonal variations in aerosol Fe solubility. For example,
desert dust aerosol mainly occurs in spring at Xi'an where our present work was conducted,
while anthropogenic emission become more important in winter (Cao and Cui, 2021);
furthermore, higher temperature in summer causes more ammonium to partition in the gas
phase and thus leads to higher aerosol acidity (Ding et al., 2019; Zhou et al., 2022). As a result,
examining seasonal variability of aerosol Fe solubility may provide clues for and insights into
factors which regulate Fe solubility. However, seasonal variation of Fe solubility has only been
explored by a few previous studies (Chen and Siefert, 2004; Tao and Murphy, 2019; Yang et
al., 2020; Yang and Weber, 2022). In the present work, we investigated seasonal variations of
total Fe, dissolved Fe and Fe solubility for fine and coarse particles at Xi'an, a megacity in
northwestern China severely affected by anthropogenic emission and desert dust (Cao and Cui,

2021).

**2 Methodology**
**2.1 Sample collection**
Aerosol sampling in Xi'an took place during 01-30 April 2021 (spring), 12 July to 14
August 2021 (summer), 07 October to 07 November 2021 (autumn) and 26 November to 31
December 2020 (winter). Xi'an has a population of ~13 million and is located in the middle of
the Guanzhong Plain which is surrounded by Qinling Mountains and Chinese Loess Plateau
(Figure S1), favoring accumulation of air pollutants and formation of severe air pollution (Cao
and Cui, 2021). In addition, Xi'an is adjacent to major deserts in China and thus frequently
affected by desert dust aerosol.
Sampling in winter took place at an urban site (34.23ºN, 108.89ºE) which is close to a
busy major road and located in residential and commercial areas (Cao et al., 2012), and was
carried out on a building roof (~10 m from the ground) in Institute of Earth Environment,
Chinese Academy of Sciences. Sampling in the other three seasons took place at another urban
site (34.37ºN, 108.97ºE) which is located in residential areas (Chen et al., 2021), and was
carried out on a building roof (~40 m from the ground) in Shaanxi University of Science and
Technology. Meteorological parameters (wind speed and direction, temperature, and relative
humidity) and $PM_{2.5}$ and $PM_{10}$ mass concentrations were provided by nearby environmental
monitoring stations.
Coarse (>1 μm) and fine (<1 μm) aerosol particles were collected onto Whatman 41 (W41)
cellulose filters on a daily basis (from 08:00 am to 07:30 am next day) using a two-stage aerosol
sampler (TH-150C, Tianhong Co., Wuhan, China) with a flow rate of 100 L/min. W41 filters,
which were used for aerosol sampling, were acid-washed to reduce background levels. After
aerosol collection, filters were sealed individually in clean plastic Petri dishes and then stored
at -20 ºC for further analysis. Our previous work (Zhang et al., 2022) described filter
pretreatment, aerosol sampling and filter storage in details. In the present work, 28, 32, 30 and
36 pairs of filter samples were collected in spring, summer, autumn and winter, respectively.
In our work, mass concentrations of various species in air, including $PM_{2.5}$ and $PM_{10}$
concentrations, are reported under standard state conditions (at 0 ºC and 1 atm) to remove the
effects of variations in temperature and atmospheric pressure.
**2.2 Sample processing and analysis**
Sample analysis was detailed in our previous work (Zhang et al., 2022; Li et al., 2023),
and as a result here we only provide key information in brief. Every filter was equally halved.
The first half filter, which was used to determine total Fe, was digested in a Teflon jar using
microwave digestion; after residual acids used in digestion were evaporated, the Teflon jar was
cooled to room temperature and then filled with 20 mL $HNO_3$ (1%). The solution was filtered
using a PTFE membrane syringe filter (pore size: 0.22 μm), and then analyzed using
inductively coupled plasma mass spectrometry (iCAP Q, Thermo Fisher Scientific, USA). In
total 14 elements were determined, including Fe, Al and Pb, and the recovery rates were found
to be 90-110% for Fe using certificated reference materials (GBW07454 and GSB07-3272-

2015).

The other half filter, which was used to determine dissolved Fe and soluble ions, was

immersed in 20 mL ultrapure water for 2 h during which an orbital shaker (300 r/min) was used
to stir the aqueous mixture. After that, the aqueous mixture was filtered using a PTFE
membrane syringe filter (pore size: 0.22 μm) and then divided further to two parts. The first
solution (~10 mL) was analyzed by ion chromatography to measure soluble anions and cations;
the second solution (10 mL) was acidified to contain 1% $HNO_3$ (using 147 μL 67% $HNO_3$) and
subsequently analyzed using inductively coupled plasma mass spectrometry.
**2.3 Aerosol acidity calculation**

The ISORROPIA-II model (Fountoukis and Nenes, 2007) was used in the "metastable +

forward" mode to calculate aerosol pH for coarse and fine particles, and input data included
concentrations of soluble anions ($SO_4^{2-}$, $NO_3^-$ and $Cl^-$) and cations ($NH_4^+$, $Na^+$, $K^+$, $Ca^{2+}$ and
$Mg^{2+}$), temperature and relative humidity (RH). The effects of $NH_3(g)$ and $HNO_3(g)$ were not
taken into account as their concentrations were not measured; this may cause some biases
(likely underestimation) in calculated aerosol pH (Guo et al., 2015; Hennigan et al., 2015; Pye
et al., 2020), but the overall trend of aerosol pH would not be significantly affected. The reverse
mode was not used in our work, as results calculated using the reverse mode are very sensitive
to uncertainties in concentrations of common aerosol ions (Hennigan et al., 2015). Coarse
particles are generally expected to be less acidic than fine particles, and it is not clear yet why
similar and even lower aerosol pH were observed for coarse particles (when compared to fine
particles) in spring and autumn at Xi'an (Figure S14). This may be caused by biases in aerosol
pH calculation, and lower aerosol pH values were also reported in previous work for coarser
particles carried out at northern Colorado, United States (Young et al., 2013). Concurrent
measurements of gaseous $NH_3$, $HNO_3$ and $HCl$ would help reduce uncertainties in calculated
aerosol pH and have been implemented in our following studies since July 2022.
**2.4 Air mass back trajectory analysis**
The Hysplit-4 model (Draxier and Hess, 1998) was employed to calculate 48-h air mass
back trajectories, using meteorological data (horizontal resolution: $1^{o} \times 1^{o}$; time resolution: 3 h)
from Global Data Assimilation System provided by National Centers for Environmental
Prediction. Back trajectories were determined with arrival height of 100 m above the ground
level and arrival time of 08:00, 14:00, 20:00 and 02:00 on the next day (Wang et al., 2020),
and every day four back trajectories were obtained. In total 120, 136, 128 and 144 back
trajectories were obtained in our work for spring, summer, autumn and winter, and all the back
trajectories were clustered using the cluster analysis method described elsewhere (Baker, 2010).
**3 Total and dissolved aerosol Fe**
**3.1 Meteorological conditions and aerosol concentrations**
The climate in Xi'an (and the Guangzhou Plain in general) is mainly regulated by the East
Asia monsoon. During our campaign, prevailing wind directions were west and southwest in
spring, northeast in summer, southwest and northeast in autumn, and west in winter (Figure
S2); furthermore, average wind speeds were >2 m/s in summer and autumn, and <2 m/s in
spring and winter. Median temperatures were 13.6, 27.0, 12.7 and 1.3 $^{\circ}$C in spring, summer,
autumn and winter, and median RH were found to be 85%, 71%, 83% and 77% (Table S1).
Precipitation mainly took place in summer during our campaign in 2021, similar to previous
years (Cao and Cui, 2021).
Table S2 shows $PM_{2.5}$ and $PM_{10}$ concentrations at Xi'an in four seasons. $PM_{10}$
concentrations were in the range of 15-243, 24-76, 22-151 and 41-212 $\mu g/m^3$ in spring, summer,
autumn and winter, and the average values were 93±61, 51±16, 70±35 and 107±39 $\mu g/m^3$,
suggesting highest levels in spring and winter and lowest levels in summer. $PM_{2.5}$ mass
concentrations were in the range of 11-62, 11-48, 13-97 and 13-156 $\mu g/m^3$, and the average
values were 35±14, 23±8, 40±24 and 80±32 $\mu g/m^3$, suggesting highest concentrations in winter
and lowest levels in summer.
$PM_{2.5}$ and $PM_{10}$ concentrations were high in winter due to accumulation of anthropogenic
pollution, and the median $PM_{2.5}/PM_{10}$ ratio was 0.76. $PM_{10}$ concentrations in spring were
significantly increased due to the impacts of desert dust aerosol, and the median $PM_{2.5}/PM_{10}$
ratio was only 0.44. Two major dust events occurred during our spring campaign (12-17 April
and 27-30 April). During the two dust events the average $PM_{10}$ and $PM_{2.5}$ mass concentrations
were 151±57 and 42±12 μg/m$^3$, and $PM_{2.5}/PM_{10}$ ratios became even lower (0.20-0.31).
Furthermore, the median $PM_{2.5}/PM_{10}$ ratios were 0.44 in summer and 0.62 in autumn.
**3.2 Total aerosol Fe**
Figure 1a shows seasonal variation of total aerosol Fe in coarse and fine particles
measured by our work at Xi'an. Total aerosol Fe concentrations were in the range of 270-3095,
191-1992, 395-3492 and 269-3924 ng/m$^3$ for coarse particles in spring, summer, autumn and
winter, and the average values were 1504±800, 950±524, 1638±830 and 1831±866 ng/m$^3$
(Table A1); total aerosol Fe concentrations were in the range of 206-12144, 164-1591, 196-
2631 and 257-4268 ng/m$^3$ for fine particles in spring, summer, autumn and winter, and the
average values were 3717±3387, 721±366, 958±516 and 2058±1037 ng/m$^3$. Average total Fe
concentrations were measured to be 798±466 and 801±534 ng/m$^3$ for coarse and fine particles
in winter (November-December 2019) at Qingdao (Zhang et al., 2022), only 44% and 38% of
the average values (1831±866 and 2058±1037 ng/m$^3$) found in winter (November-December
2020) at Xi'an by the present work, mainly because during wintertime aerosol mass
concentrations were much higher at Xi'an than Qingdao.
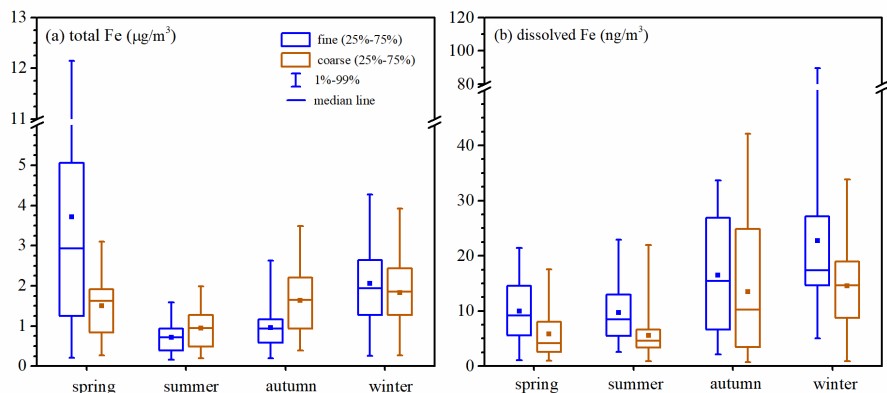

**Figure 1.** Seasonal variations of (a) total Fe and (b) dissolved Fe for fine and coarse particles.

The average contribution of coarse particles to total Fe in TSP (total suspended particles)
were 29%, 57%, 63% and 47% in spring, summer, autumn and winter, being lowest in spring
when the influence of desert dust aerosol was largest. Statistical analysis (paired t-test)
suggested that compared to fine particles, total Fe in coarse particles was significantly lower in
spring ($p<0.01$, $\alpha=0.05$) while significantly higher in summer and autumn ($p<0.01$, $\alpha=0.05$); in
addition, there was no significant difference between coarse and fine particles in winter ($p=0.13$,
$\alpha=0.05$). The average contribution of coarse particles to total Al in TSP were 26%, 50%, 54%
and 40% in spring, summer, autumn and winter, also being lowest in spring. Furthermore,
during dust periods (12-17 and 27-30 April) the average contribution of coarse particles was
found to be 24% for total Fe and 23% for total Al in TSP, even slightly smaller than the average
values in spring. This is probably because for desert dust aerosol, Fe and Al were enriched in
fine particles while the major component in coarse particles was quartz (Journet et al., 2014);
nevertheless, further measurements are needed to better understand size distribution of trace
metals in desert dust aerosol.
Compared to other seasons, total Fe in fine and coarse particles were both lowest in
summer (Figure 1a), as aerosol mass concentrations were also lowest in summer (Section 3.1).
Similarly, previous measurements on Huaniao Island in the East China Sea (Yang et al., 2020)
and over the tropical and subtropical North Atlantic (Chen and Siefert, 2004) also found lowest
total aerosol Fe levels in summer. For the other three seasons (spring, autumn and winter), total
Fe in coarse particles were rather similar, while total Fe in fine particles were highest in spring
and lowest in autumn. Overall, compared to summer and autumn, total aerosol Fe were higher
in spring and winter when higher aerosol mass concentrations were also observed (Figure S3).
Total Fe was very well correlated with total Al ($0.87 < r < 0.96$, $p < 0.01$) for both coarse
(Figure S4) and fine particles (Figure 2) in all the four seasons, suggesting desert dust always
as the dominant source for total aerosol Fe at Xi'an, regardless of particle size range and
seasons. The median Fe/Al values, mass ratios of total Fe to total Al, were 0.975, 0.926, 1.269
and 0.940 in spring, summer, autumn and winter for coarse particles, and 0.735, 0.796, 0.870
and 0.744 for fine particles (Figure S5). Fe/Al were found to be 0.911 and 0.741 for $PM_{10}$ and
$PM_{2.5}$ generated using surface soil samples collected over several major deserts in China
(Zhang et al., 2014). We found that Fe/Al measured for coarse and fine particles at Xi'an were
similar to these reported for desert dust (Zhang et al., 2014); coarse particles in autumn might
be one exception (Figure S5), showing slightly higher Fe/Al (median: 1.269) than desert dust.

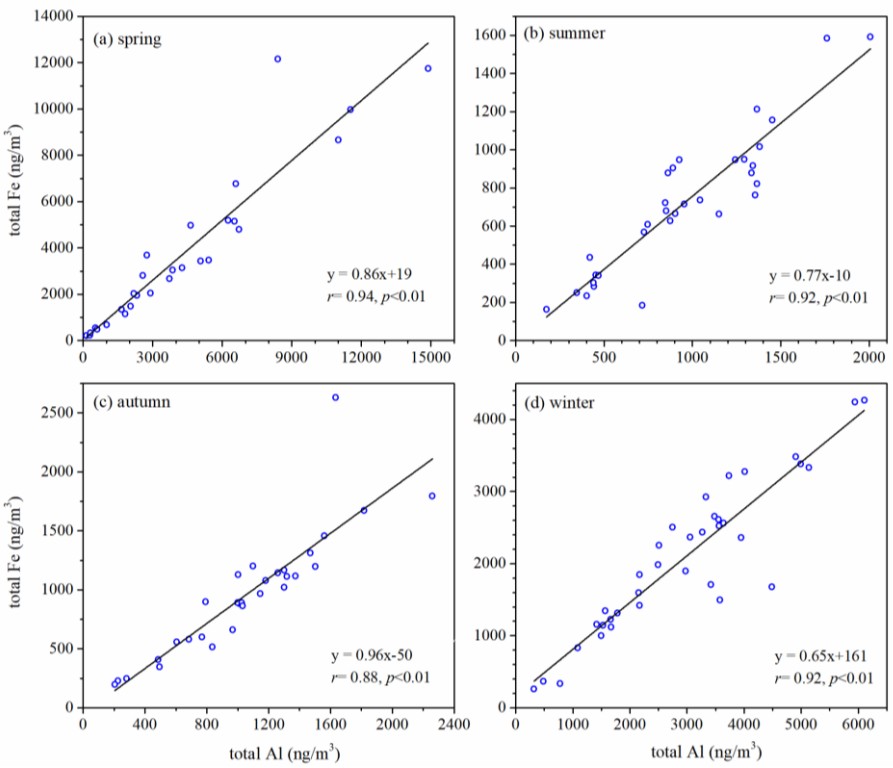

**Figure 2.** Total Fe versus total Al for fine particles in different seasons: (a) spring; (b) summer; (c) autumn; (d) winter.

### 3.3 Dissolved aerosol Fe

Figure 1b shows seasonal variation of dissolved aerosol Fe in coarse and fine particles at Xi'an. Dissolved aerosol Fe concentrations were in the range of 1.0-17.5, 0.9-22.0, 0.7-42.2 and 0.9-33.8 ng/m$^3$ for coarse particles in spring, summer, autumn and winter, and the average values were 5.9±4.5, 5.6±4.0, 13.5±12.2 and 14.5±8.3 ng/m$^3$; dissolved aerosol Fe concentrations were in the range of 1.1-21.4, 2.6-22.9, 2.1-33.7 and 5.0-89.5 ng/m$^3$ for fine particles in spring, summer, autumn and winter, and the average values were 10.0±5.5, 9.7±5.6, 16.5±10.1 and 22.7±16.8 ng/m$^3$. Average dissolved Fe concentrations were measured to be 7.7±14.5 and 7.3±7.6 ng/m$^3$ for coarse and fine particles in winter at Qingdao (Zhang et al.,

2022), only 53% and 32% of the average values (14.5±8.3 and 22.7±16.8 ng/m$^3$) found in
winter at Xi'an by the present work, and one major reason is that total Fe concentrations were
significantly higher at Xi'an than Qingdao.
The average contribution of coarse particles to dissolved Fe in TSP were 37%, 36%, 45%
and 39% in spring, summer, autumn and winter. Compared to fine particles, dissolved Fe was
significantly lower in coarse particles for all the four seasons (paired t-test, p<0.01, α=0.05) at
Xi'an, although total Fe in coarse particles were higher than or similar to fine particles (except
spring, as discussed in Section 3.2). This indicated that aerosol Fe solubility was lower in
coarse particle than fine particles, as further discussed in Section 4. Similar to our work, Sakata
et al. (2022) found that over the Pacific dissolved aerosol Fe concentrations in fine particles
(<1.3 μm) were significantly higher than coarse particles (>1.3 μm).
Compared to spring and summer, dissolved Fe concentrations were higher in autumn and
winter for coarse particles (Figure 1b); for fine particles, dissolved Fe concentrations were
highest in winter, followed by autumn, and lowest in spring and summer. Dissolved Fe
concentrations were low in summer, as total Fe concentrations were also low (Figure 1a). Total
Fe concentrations were high in spring (Figure 1a), but dissolved Fe concentrations were low;
this is because compared to other seasons, spring was most severely affected by desert dust
with low Fe solubility. Our previous study (Zhang et al., 2022) investigated aerosol Fe at
Qingdao in winter, and found that compared to clean days, dissolved Fe concentrations did not
change significantly during dust days although total Fe concentrations were remarkably
increased. Therefore, our previous (Zhang et al., 2022) and current studies imply that the
occurrence of desert dust aerosol may not necessarily lead to increase in dissolved Fe
concentrations in the air.

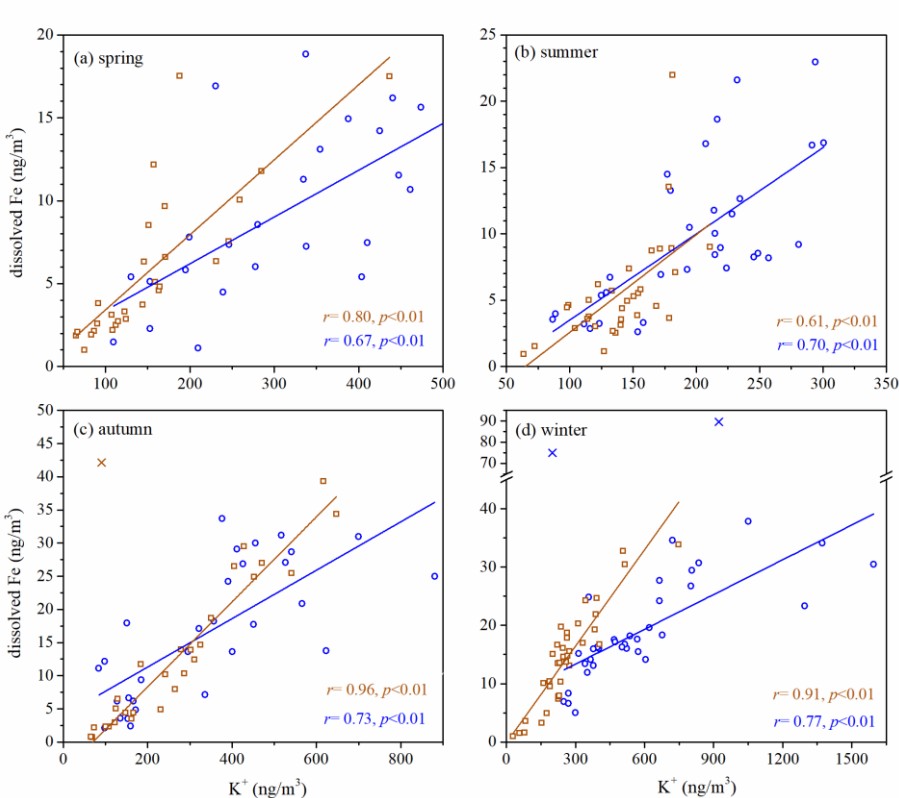

**Figure 3.** Dissolved Fe versus $K^+$ for fine and coarse particles in different seasons: (a) spring;
(b) summer; (c) autumn; (d) winter. Blue symbols represent fine particles and brown symbols
represent coarse particles. Cross symbols represent data points which are not included in
fittings.

As shown in Figure S6, overall the correlation between dissolved Fe and total Al was quite
weak at Xi'an, indicating that desert dust may not contribute dominantly to dissolved aerosol
Fe, although it was always the major source of total aerosol Fe (Section 3.2). We also examined
correlations between dissolved Fe and several other species (Table S3). Except for summer,

dissolved Fe was well correlated with secondary inorganic species (sulfate, nitrate and ammonium) for coarse and fine particles, suggesting secondary formation (i.e. conversion of insoluble Fe to dissolved Fe via chemical processing) as an important source of dissolved Fe. Besides, Figure 3 shows that dissolved Fe was well correlated with $K^+$ (a tracer for biomass burning) in coarse and fine particles at all the four seasons ($0.67<r<0.96$, $p<0.01$), and this may indicate biomass burning also as an important source for dissolved aerosol Fe.

Furthermore, good correlations with dissolved Fe were found in coarse particles for Pb, Zn and As in three seasons (spring, autumn and winter), and in fine particles for Pb (spring, autumn and winter), As (spring and autumn) and Zn (autumn). Aerosol Pb and Zn are mainly emitted by vehicles and iron-steel industry (Chow et al., 2004; Cao and Cui, 2021), and the major sources of aerosol As include coal combustion and metal smelting (Tian et al., 2010). As a result, vehicle emission, coal combustion, iron-steel industry and metal smelting also contributed to dissolved aerosol Fe at Xi'an.

## 4 Aerosol Fe solubility

### 4.1 Seasonal variation of Fe solubility

### 4.1.1 Seasonal variability

Figure 4 and Table A1 display aerosol Fe solubility at Xi'an in different seasons. Fe solubility was in the range of 0.08-2.48%, 0.13-2.44%, 0.05-3.55% and 0.09-7.16% for coarse particles in spring, summer, autumn and winter, and the median values were 0.38%, 0.51%, 0.62% and 0.79%; for fine particles, Fe solubility was in the range of 0.06-1.26%, 0.34-3.02%,

0.27-3.37% and 0.21-9.65% in spring, summer, autumn and winter, and the median values
were 0.42%, 1.35%, 1.79% and 1.17%.

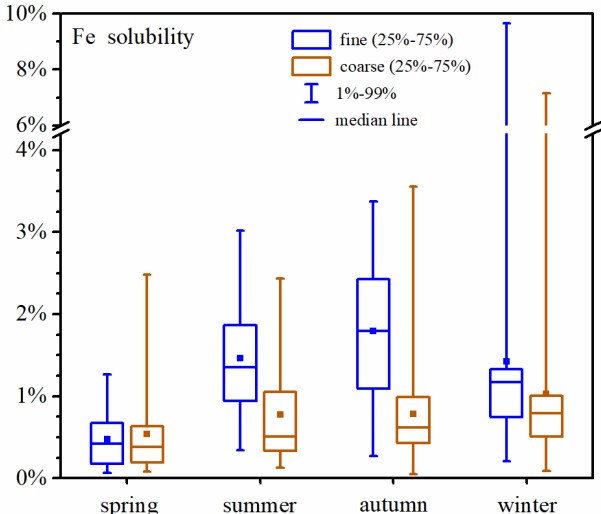


**Figure 4.** Seasonal variations of aerosol Fe solubility for fine and coarse particles at Xi'an.

No significant difference in Fe solubility was found between coarse and fine particles at
Xi'an in spring (paired *t*-test, p=0.17, α=0.05). In addition, the median values of Fe solubility
were both <0.5% for coarse and fine particles in spring, similar to desert dust (Schroth et al.,
2009; Shi et al., 2011b; Oakes et al., 2012b; Li et al., 2022), and this was because Xi'an was
frequently affected by desert dust aerosol in spring. In the other three seasons (summer, autumn
and winter), Fe solubility was significantly higher in fine particles than coarse particles (paired
*t*-test, p<0.01, α=0.05); furthermore, in these three seasons the median Fe solubility was >1%
for fine particles and >0.5% for coarse particles. For coarse particles, Fe solubility was highest
in winter and lowest in spring, while no significant difference was found between summer and
autumn (t-test, p=0.95, α=0.05); for fine particles, Fe solubility can be described by the
following order: autumn > summer > winter > spring.

A number of field measurements (Hsu et al., 2005; Baker and Jickells, 2006; Sedwick et

al., 2007; Kumar et al., 2010; Sholkovitz et al., 2012; Winton et al., 2015; Shelley et al., 2018;
Yang et al., 2023) found inverse dependence of Fe solubility on total Fe (and Al). As shown in
Figures 5 and S7, Fe solubility was also observed in our work to decrease with total Fe for
coarse and fine particles in three seasons (spring, summer and winter), and such dependence
can be fitted using Eq. (1):

$$f_s(Fe) = a \times [Fe]_T^{-b} \qquad (1)$$

where $f_s$(Fe) is Fe solubility and $[Fe]_T$ is total Fe concentration, and $b$ represents the sensitivity
of Fe solubility to relative change in total Fe concentration. As shown in Figures S8-S9, such
inverse dependence was also observed between Fe solubility and total Al in these three seasons.
Several mechanisms can qualitatively explain such inverse dependence, but a consensus has
not been reached yet (Mahowald et al., 2018; Meskhidze et al., 2019).

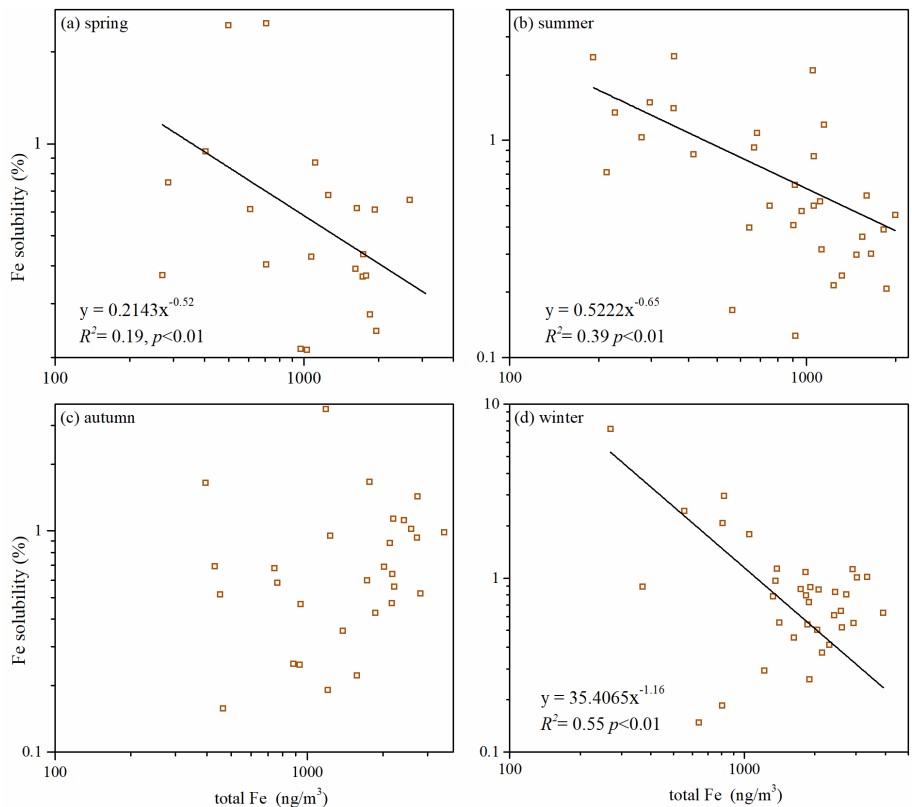


**Figure 5.** Fe solubility versus total Fe for coarse particles in different seasons: (a) spring; (b)

summer; (c) autumn; (d) winter.

However, no obvious relationship between Fe solubility and total Fe (or total Al) was

found in autumn. Such inverse dependence was not found in some previous studies either (Paris

et al., 2010; Oakes et al., 2012a), and was found for fine particles but not for coarse particles

at Qingdao in the winter by our previous work (Zhang et al., 2022). Therefore, one may

conclude that the inverse dependence of Fe solubility on total Fe (or total Al), though frequently

observed, is not a universe rule. It is not clear yet why such inverse dependence was not

observed in these studies.

A larger *b* value means that Fe solubility is more sensitive to relative change in total Fe
concentration. For our measurements conducted at Xi'an, *b* values were determined to be 0.30,
0.23 and 0.91 in spring, summer and winter for fine particles (Figure S7), and 0.52, 0.65 and
1.16 for coarse particles (Figure 5). One can see that the *b* values in winter were much larger
than those in spring and summer for both fine and coarse particles; furthermore, in each of the
three seasons (spring, summer and winter), the *b* value was larger for coarse particles than fine
particles.
**4.1.2 Comparison with previous work**
The median Fe solubility at Xi'an were 0.79% and 1.17% for coarse and fine particles in
winter, larger than those (0.34% and 0.66%, respectively) found in winter at Qingdao (Zhang
et al., 2022). One reason is that in winter Xi'an was frequently affected by haze pollution when
aerosol Fe solubility was significantly increased (Shi et al., 2020; Zhang et al., 2022; Zhu et
al., 2022); in addition, our winter sampling at Qingdao was severely affected by desert dust
aerosol (Zhang et al., 2022) with very low Fe solubility.
Several previous studies also investigated aerosol Fe solubility in northern China. Similar
to our current work, Chuang et al. (2005) reported low Fe solubility (<1%) for TSP during
spring at Dunhuang, a city in Northwest China. Average Fe solubility at Xi'an were reported
to be 10.4% for TSP (He et al., 2021) and 25.5±11.3% for $PM_{2.5}$ (Lei et al., 2023), 5.0±3.8%,
4.5±2.6% and 2.7±1.5% for $PM_{2.5}$ at three cities in North China (Zhu et al., 2020), and
2.70±2.77% for TSP at Qingdao (Shi et al., 2020). Compared to our work, some other studies
(Shi et al., 2020; Zhu et al., 2020; He et al., 2021; Lei et al., 2023) reported higher Fe solubility,
mainly because different leaching protocols were employed to extract dissolved Fe (Meskhidze
et al., 2019; Li et al., 2023): 1) sonication was used during extraction in three previous studies
(Shi et al., 2020; Zhu et al., 2020; He et al., 2021) but not in our work; 2) filter pore size Shi et
al. used (0.45 μm) was larger than that our work used (0.22 μm); 3) the leaching solution used
by Lei et al. (acetate buffer, pH=4.3) was more acidic than that we used (ultrapure water).

**4.1.3 Fe solubility during dust and haze events**

At Xi'an, dust events ($PM_{10}$ >100 μg/m$^3$ and $PM_{10}$/$PM_{2.5}$ >3) occurred in spring (12-17
and 27-30 April). During the two dust events average Fe solubility was 0.17±0.09% and
0.18±0.13% for coarse and fine particles, similar to that reported for dust particles collected
from dust source regions (Shi et al., 2011b; Oakes et al., 2012b; Paris and Desboeufs, 2013; Li
et al., 2022); during non-dust periods in spring, average Fe solubility was found to be 0.75±0.66%
and 0.64±0.27% for coarse and fine particles, higher than that observed for dust events. In fact,
much lower Fe solubility was also reported during dust events at Qingdao (Shi et al., 2020;
Zhang et al., 2022), Jeju Island (Chuang et al., 2005) and Hokkaido (Ooki et al., 2009), when
compared to non-dust periods.
In this work we classified high-RH haze events as those with $PM_{2.5}$>80 μg/m$^3$,
$PM_{2.5}$/$PM_{10}$>0.8 and RH>80%. High-RH haze events at Xi'an only occurred in winter (26
November to 01 December, and 05-07 December). Average Fe solubility was measured to be
2.03±2.07% and 2.16±2.81% for coarse and fine particles during high-RH haze events,
significantly higher than that observed for other days in winter (0.69±0.46% and 1.18±0.81%
on average, respectively). Some previous studies (Shi et al., 2020; Zhu et al., 2020; Zhang et
al., 2022; Zhu et al., 2022) also observed evaluated aerosol Fe solubility during haze periods,
attributed to increased contribution of anthropogenic Fe with high solubility and/or Fe
solubility enhancement via chemical processing.

**4.2 Influence of air mass sources on Fe solubility**

Back trajectories obtained for our campaign were clustered, and we further examined the

dependence of Fe solubility on air mass cluster types in different seasons. In spring (Figure 6a),
air mass culster C1 originated locally and C2 originated from North China Plain with severe
air pollution, while C3 and C4 represented air mass arriving from desert regions in the north
and northwest after long-distance transport (compared to C1 and C2); as shown in Figure 6b,
Fe solubility in coarse and fine particles was significantly higher for C1 and C2, when
compared to C3 and C4.

In autumn (Figure 6e), air mass cluster C1 originated locally and C2 was transported from

desert regions in the north/northwest, and Fe solubility was much higher for C1 than C2 (Figure
6f). In winter (Figure 6g), air mass cluster C1 originated locally while C2, C3 and C4 originated
from desert regions in the north and northwest, and the transport distance increased from C2 to
C4; as shown in Figure 6h, Fe solubility followed the order C1 > C2 > C3 > C4, decreasing
with increase in transport distance. In contrast to other three seasons, no obvious dependence
of Fe solubility on air mass clusters was found in summer (Figure 6d).

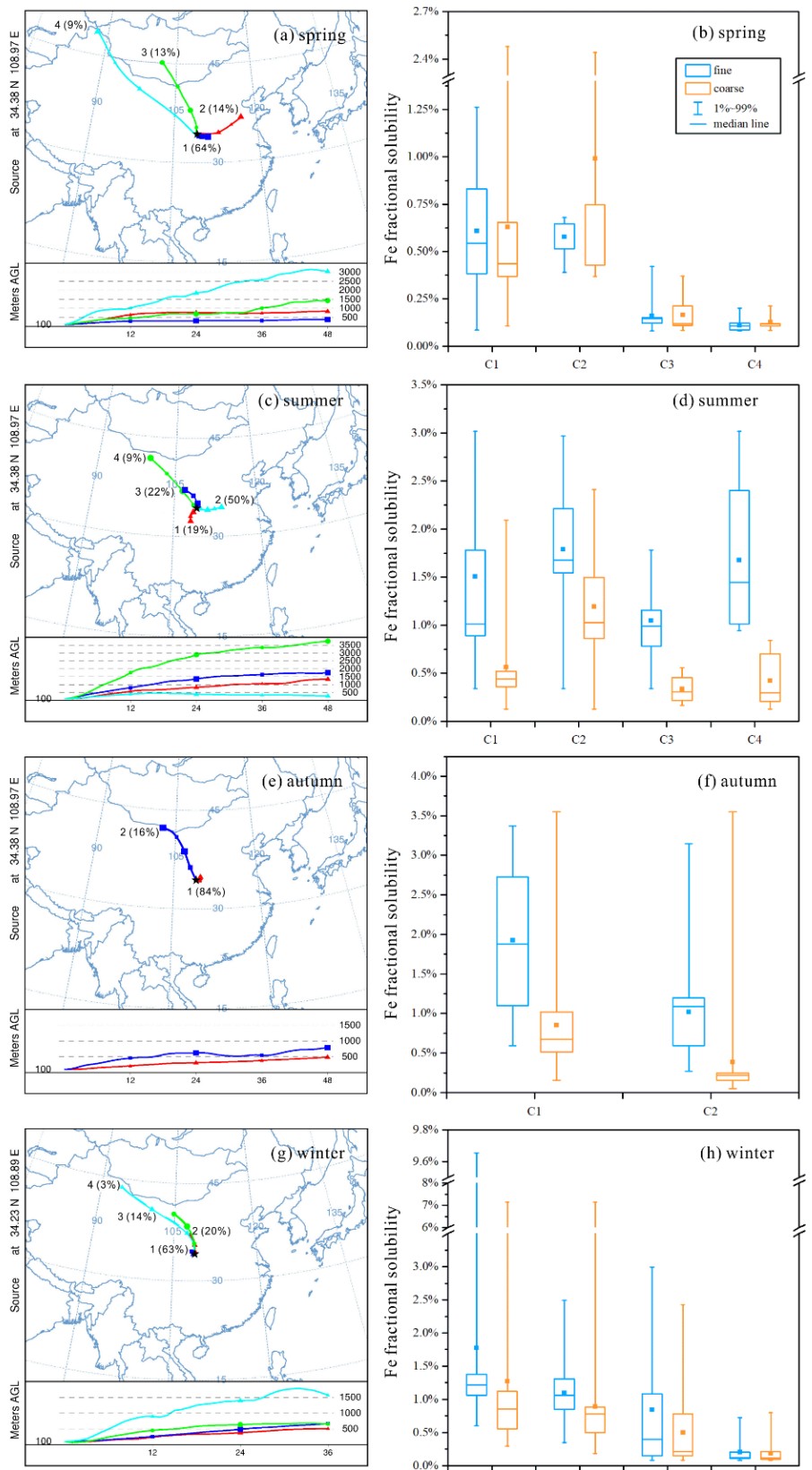


**Figure 6.** The mean backward trajectory clusters obtained by HYSPLIT for (a) spring, (c)
summer, (e) autumn, and (g) winter; Fe solubility in fine and coarse particles for different air
mass clusters in (b) spring, (d) summer, (f) autumn, and (h) winter. C1-C4 represent different
air mass clusters.

To summarize, our work found that in spring, autumn and winter, Fe solubility at Xi'an

was significantly higher when air masses originated from local and nearby regions, when
compared to those arriving from desert regions after long-distance transport. The reason is that
the contriution of anthropogenic emissions to aerosol Fe was elevated for air masses originating
from local and nearby sources (when compared to air masses originating from desert regions),
and anthropogenic aerosol Fe had higher solubility than desert dust (Schroth et al., 2009; Fu et
al., 2012; Oakes et al., 2012b). Simialr to our work, over the Sargasso Sea aerosol Fe solubility
was much lower in Saharan air masses than North American air masses (Sedwick et al., 2007).
**4.3 Effects of chemical aging**

Laboratory studies (Shi et al., 2011a; Chen and Grassian, 2013; Wang et al., 2018)

suggested that chemical processing by acids, such as $H_2SO_4$ and $HNO_3$, could dissolve
insoluble Fe and thus enhance aerosol Fe solubility. Some field studies found that aerosol Fe
solubility was positively correlated with sulfate and/or nitrate (Shi et al., 2020; Zhu et al., 2020;
Liu et al., 2021; Zhang et al., 2022; Yang et al., 2023), indicating enhancement of Fe solubility
by atmospheric acid processing.

Figure 7 plots Fe solubility at Xi'an versus (2×[sulfate]+[nitrate])/[Fe] (in nmol/nmol,

referred to as relative abundance of aerosol acidic species), the molar ratio of two major acidic
species to total Fe in aerosol particles. For coarse particles, aerosol Fe solubility was well
correlated with relative abundance of aerosol acidic species in all the four seasons (0.77 <$r$
<0.91, p<0.01). For fine particles, good correlation was found in spring ($r$=0.84, p<0.01),
moderate correlation was found in autumn and winter (0.42<$r$<0.53, p<0.01), and no
significant correlation was found in summer ($r$=0.35, p>0.05). In addition, as shown in Figures
S10-S11, correlations of Fe solubility with [nitrate]/[Fe] were better than (or very similar to)
these with [sulfate]/[Fe]for coarse particles in the four seasons, whereas no obvious trend was
not observed for fine particles.

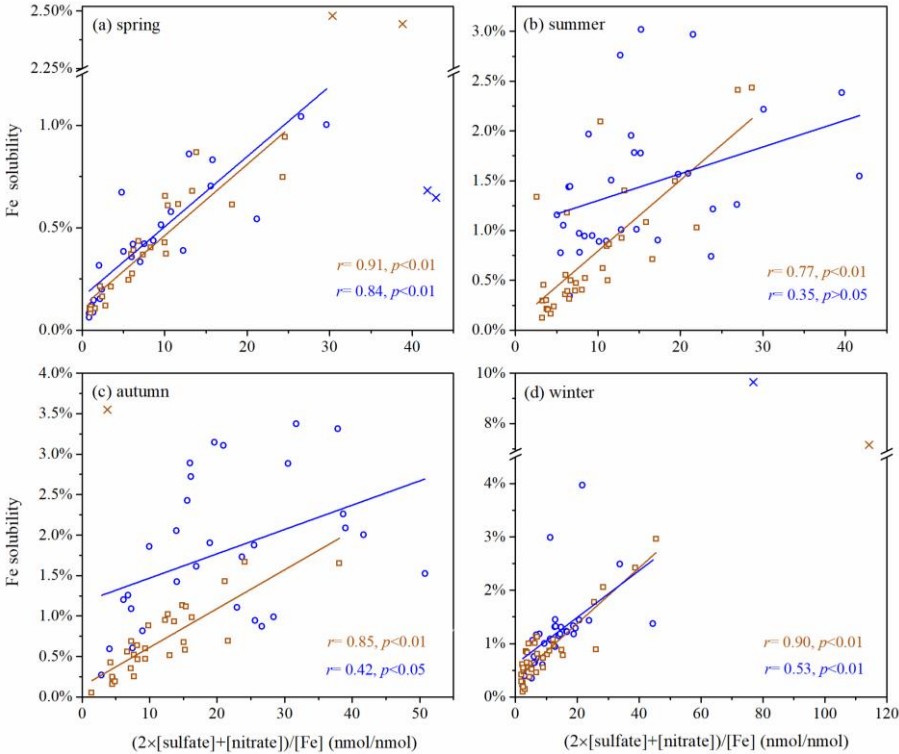


**Figure 7.** Fe solubility versus (2×[sulfate]+[nitrate])/[Fe] for fine and coarse particles in
different seasons: (a) spring; (b) summer; (c) autumn; (d) winter. Blue symbols represent fine
particles and brown symbols represent coarse particles. Cross symbols represent data points
which are not included in fittings.

Overall, correlations between Fe solubility and relative abundance of aerosol acidic
species were always better for coarse particles than fine particles (Figure 7), indicating that
acid processing may be more important in Fe solubility enhancement for coarse particles, when
compared to fine particles. A previous study (Zhang et al., 2022) also found that such
correlation was better in coarse particles than fine particles in winter at Qingdao, a coastal city
in northern China. Nevertheless, as discussed in Section 4.1, Fe solubility was higher in fine
particles than coarse particles. This may imply that primary emission of non-desert-dust Fe
(anthropogenic Fe) with higher solubility (Schroth et al., 2009; Oakes et al., 2012b) was more
important for Fe solubility enhancement in fine particles than coarse particles.
It was suggested by laboratory studies (Chen and Grassian, 2013; Paris and Desboeufs,
2013; Wang et al., 2017) that atmospheric organic ligands, such as oxalate, could increase
aerosol Fe solubility via ligand-promoted dissolution. As shown in Figure S12, our present
work found good correlation between Fe solubility with [oxalate]/[Fe] (in nmol/nmol) for
coarse particles ($0.70 < r < 0.88$, $p < 0.01$) and moderate correlation for fine particles ($0.40 < r < 0.67$,
$p < 0.01$) at Xi'an. Positive correlation between Fe solubility and oxalate was also observed
previously at Atlanta, USA (Yang and Weber, 2022), Toronto, Canada (Tao et al., 2022) and
Qingdao, China (Zhang et al., 2022).
We note that good correlation between Fe solubility and aerosol oxalate does not
necessarily mean Fe solubility enhancement by Fe-oxalate complexation. For example, it was
suggested that Fe could promote the formation of oxalate in aerosol particles (Tao and Murphy,
2019; Zhang et al., 2019), and thus good correlation between Fe solubility and oxalate could
also imply enhanced formation of oxalate by dissolved Fe. In addition, similar to sulfate and
nitrate, the major source of oxalate in the troposphere was secondary formation
(Myriokefalitakis et al., 2011; Kawamura and Bikkina, 2016), and in this aspect good
correlation between Fe solubility and relative abundance of oxalate could also indicate the
importance of secondary formation of dissolved aerosol Fe (i.e. dissolution of insoluble Fe to
dissolved Fe via aging processes).
**5 Discussion: roles of RH and aerosol acidity**
Figure 8 reveals the importance of RH in regulating aerosol Fe solubility. When RH was
increased from <60% to 60-70%, significant increase in Fe solubility was observed for both
coarse and fine particles. Sun et al. (2018) investigated hygroscopicity of aerosol particles
collected in North China, and found that most particles examined started to become deliquesced
when RH was increased to ~60%. The deliquescence RH reported for ambient aerosol particles
(Sun et al., 2018) coincided roughly with the RH threshold at which large increase in aerosol
Fe solubility was observed in our work. Previous studies (Shi et al., 2020; Zhu et al., 2020; Zhu
et al., 2022) also highlighted that RH and thus aerosol liquid water could substantially affect
Fe solubility. For examples, Zhu et al. (2020) measured Fe solubility at four cities in eastern

China in December 2017, and found that Fe solubility at >50% RH was significantly larger

than that at <50% RH.

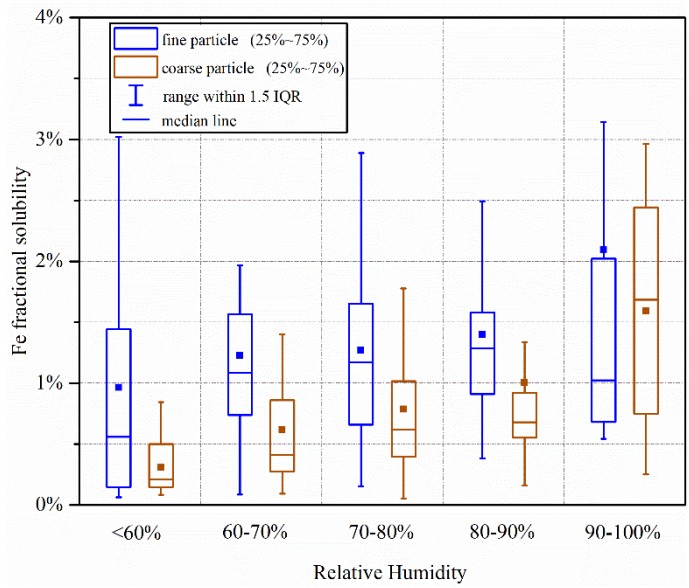

**Figure 8.** Fe solubility in different RH (relative humidity) ranges for fine and coarse particles.

(RH<60%: 18 days; 60%<RH<70%: 23 days; 70%<RH<80%: 48 days; 80%<RH<90%: 28

days; RH>90%: 10 days).

In addition, as shown in Figure 8, when RH was increased from 80-90% to >90%, median

Fe solubility was remarkably increased from 0.67% to 1.68% for coarse particles. Similar to

our work, Shi et al. (2020) also found that aerosol Fe solubility at Qingdao was significantly

increased under foggy weather when compared to other weather conditions. Therefore, both

Shi et al. (2020) and our present work suggested that high RH could promote Fe dissolution.

We further examined the impact of aerosol acidity on Fe solubility, and the results are

displayed in Figure 9. For coarse particles, increase in pH did not lead to apparent change in

Fe solubility as long as aerosol pH was <5; however, Fe solubility was greatly decreased when

aerosol pH was increased to >5. For fine particles, Fe solubility in general decreased with
increasing aerosol pH (from <2 to >5). Previous work carried out at six Canadian sites (Tao
and Murphy, 2019) and Atlanta, USA (Wong et al., 2020; Yang and Weber, 2022) also reported
higher Fe solubility at lower aerosol pH. Similar to our previous work at Qingdao in the winter
(Zhang et al., 2022), our current study found that for coarse and fine particles at Xi'an, aerosol
pH was mostly <4 when Fe solubility exceeded 1% (Figure S13). It should be pointed out that
for some samples collected at Xi'an, Fe solubility could still be very low (<1%) even when
aerosol pH was low and RH was high (Figure S13). In total 34 samples for coarse particles (9
in spring, 8 in summer, 12 in autumn and 5 in winter) and 18 samples for fine particles (7 in
spring, 6 in summer, 4 in autumn and 1 in winter) fulfilled the above conditions (pH<4,
RH>80%, and Fe solubility <1%). Fe mineralogy may possibly explain the observed low Fe
solubility despite high RH and aerosol acidity, and concurrent measurements of Fe mineralogy
could provide further clues.

In addition, as shown in Figure 9, at a given pH range Fe solubility was always higher in

fine particles than coarse particles. If we assume at the same pH range Fe solubility
enhancement by acid processing was similar for fine and coarse particles, the results displayed
in Figure 9 may imply that anthropogenic and pyrogenic Fe played a more important role in Fe
solubility enhancement in fine particles at Xi'an, when compared to coarse particles. Mcdaniel
et al. (2019) found that soluble Fe concentration was strongly correlated with aerosol surface
area for size-resolved aerosol samples collected from several different regions, and suggested
surface area as the main factor which affected Fe solubility; they further suggested that this
was because Fe solubility enhancement by acid processing could be more effective for aerosol
particles with larger surface area and thus smaller particle size.

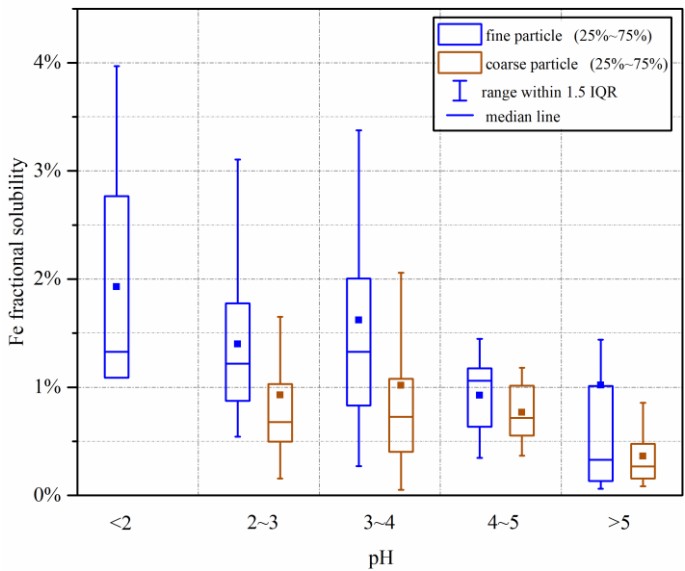


**Figure 9.** Fe solubility in different pH ranges for fine and coarse particles at Xi'an.

Our work found that at Xi'an aerosol pH values for both coarse and fine particles were

lower (*t*-test, p<0.01, α=0.05) in summer and autumn than spring and winter (Table S4 and
Figure S14). Compared to summer and autumn, lower temperature in winter favored
partitioning of ammonium in aerosol particles and thus led to higher aerosol pH. Average
temperatures were similar in spring and autumn at Xi'an (Table S1), but aerosol pH was higher
in spring than autumn (Table S4). Higher aerosol pH in spring at Xi'an, when compared to
autumn, was caused by increase of non-volatile cations in spring due to the influence of desert
dust aerosol; in fact, we found that the abundance of $Ca^{2+}$ (relative to sulfate) was much higher
in spring for both fine and coarse particles. Meanwhile, Fe solubility was higher in summer
(median: 1.35%) and autumn (median: 1.79%) than spring (median: 0.42%) and winter (median:
1.17%) for fine particles, and was also higher in summer (median: 0.51%) and autumn (median:
0.62%) than spring (median: 0.38%) for coarse particles. As a result, lower aerosol pH (thus
higher aerosol acidity) in summer and autumn may at least partly explain the observed higher
Fe solubility in these two seasons. Our results were corroborated by a previous study (Yang
and Weber, 2022) which found that compared to the cold season, higher Fe solubility was
found at Atlanta (Georgia, USA) in the warm season when aerosol pH was lower.

## 6 Summary and conclusion

Our work investigated total Fe, dissolved Fe and Fe solubility for coarse (>1 μm) and fine
(<1 μm) particles in four different seasons at Xi'an, a megacity in northwestern China impacted
by anthropogenic emissions and desert dust. Total Fe concentrations in coarse particles were
lowest in summer and similar in the other three seasons, while for fine particles total Fe
concentrations were lowest in summer and highest in spring. Good correlations were found
between total Fe and total Al for both coarse and fine particles in all the four seasons,
suggesting desert dust aerosol as the major source of total Fe regardless of particle size (below
or above 1 μm) and season.
Dissolved Fe concentrations were higher in autumn and winter than spring and summer
for coarse particles; for fine particles, dissolved Fe concentrations were highest in winter,
followed by autumn, and lowest in spring and summer. Compared to other seasons, although
total Fe concentrations were evaluated in spring due to the impacts of desert dust, increase in
dissolved Fe levels was not observed. This may imply that the occurrence of desert dust aerosol
may not necessarily lead to increase in dissolved Fe concentrations, as also revealed in our
previous study (Zhang et al., 2022) carried out at a coastal city in northern China. Dissolved
Fe was significantly lower for coarse particles (compared to fine particles) in all the four
seasons, although total Fe in coarse particles were higher than or similar to fine particles in
three seasons (but not spring), implying higher Fe solubility in fine particles. Overall the
correlation between dissolved Fe and total Al was rather weak, suggesting that desert dust may
not contribute dominantly to dissolved Fe at Xi'an, although it was always the major source of
total Fe.

Highest Fe solubility was observed in winter for coarse particles and in autumn for fine

particles; meanwhile, lowest Fe solubility was observed in spring for both coarse and fine
particles, with median Fe solubility both below 0.5%. Compared to coarse particles, Fe
solubility was similar for fine particles in spring but significantly higher in the other three
seasons. Inverse dependence of Fe solubility on total Fe concentration was observed for coarse
and fine particles in spring, summer and winter, while there was no such dependence for either
fine or coarse particles in autumn. Furthermore, aerosol Fe solubility was higher in air masses
originating from local and nearby regions than those arriving from desert regions after long-
distance transport in three seasons (spring, autumn and winter), while no apparent dependence
of Fe solubility on air mass origins was found in summer.

Our work found better correlation between Fe solubility and relative abundance of aerosol

acidic species for coarse particles than fine particles in all the four seasons, probably suggesting
that acid processing was more important for Fe solubility enhancement in coarse particles. This
may further mean that non-desert-dust Fe (e.g., anthropogenic and biomass burning Fe) was
more important for Fe solubility enhancement in fine particles, since Fe solubility was higher
in fine particles than coarse particles. We also found that overall Fe solubility increased with
RH and acid acidity for coarse and fine particles, underscoring the importance of aerosol liquid
water and aerosol acidity in enhancing Fe solubility via acid processing. Our work further
found that at a given pH range aerosol Fe solubility was always higher in fine particles than
coarse particles.


## Appendices

**Table A1.** Overview of total Fe (in ng/m$^3$), dissolved Fe (in ng/m$^3$) and Fe solubility (in %) for

fine and coarse particles in different seasons at Xi'an.

| | fine particles | | | coarse particles | | |
|---|---|---|---|---|---|---|
| **Spring** | **range** | **median** | **average** | **range** | **median** | **average** |
| total Fe | 206-12144 | 2925 | 3717±3387 | 270-3095 | 1626 | 1504±800 |
| dissolved Fe | 1.1-21.4 | 9.2 | 10.0±5.5 | 1.0-17.5 | 4.2 | 5.9±4.5 |
| Fe solubility | 0.06-1.26 | 0.42 | 0.48±0.32 | 0.08-2.48 | 0.38 | 0.54±0.59 |
| **Summer** | **range** | **median** | **average** | **range** | **median** | **average** |
| total Fe | 164-1591 | 719 | 721±366 | 191-1992 | 942 | 950±524 |
| dissolved Fe | 2.6-22.9 | 8.5 | 9.7±5.6 | 0.9-22.0 | 4.6 | 5.6±4.0 |
| Fe solubility | 0.34-3.02 | 1.35 | 1.46±0.67 | 0.13-2.44 | 0.51 | 0.78±0.63 |
| **Autumn** | **range** | **median** | **average** | **range** | **median** | **average** |
| total Fe | 196-2631 | 934 | 958±516 | 395-3492 | 1651 | 1638±830 |
| dissolved Fe | 2.1-33.7 | 15.4 | 16.5±10.1 | 0.7-42.2 | 10.3 | 13.5±12.2 |
| Fe solubility | 0.27-3.37 | 1.79 | 1.80±0.88 | 0.05-3.55 | 0.62 | 0.79±0.67 |
| **Winter** | **range** | **median** | **average** | **range** | **median** | **average** |
| total Fe | 257-4268 | 1942 | 2058±1037 | 269-3924 | 1850 | 1831±866 |
| dissolved Fe | 5.0-89.5 | 17.4 | 22.7±16.8 | 0.9-33.8 | 14.7 | 14.5±8.3 |
| Fe solubility | 0.21-9.65 | 1.17 | 1.43±1.58 | 0.09-7.16 | 0.79 | 1.03±1.22 |

**Data availability.**

Data are available upon request (Mingjin Tang: mingjintang@gig.ac.cn).

**Competing interests.**

The authors declare that they have no conflict of interest.

**Author contribution.**

**Huanhuan Zhang:** investigation, formal analysis, writing-original draft, writing-review &
editing; **Rui Li:** investigation, writing-original draft; **Chengpeng Huang**: investigation;
**Xiaofei Li**: investigation; **Shuwei Dong:** investigation; **Fu Wang:** investigation; **Tingting Li:**
investigation; **Yizhu Chen:** investigation; **Guohua Zhang:** resource, writing-review & editing;
**Yan Ren:** resource; **Qingcai Chen:** resource; **Ru-jin Huang:** resource; **Siyu Chen:** writing-
review & editing; **Xinming Wang:** resource; **Mingjin Tang:** conceptualization, formal
analysis, writing-original draft, writing-review & editing.
**Financial support.**
This work was sponsored by National Natural Science Foundation of China (42022050 and
42277088), China Postdoctoral Science Foundation (2021M703222), Guangdong Foundation
for Program of Science and Technology Research (2019B121205006 and 2020B1212060053),
Guangdong Province (2017GC010501) and the CAS Pioneer Hundred Talents program.
**Acknowledgement.**
We would like to thank Dr. Shiguo Jia at Sun Yat-sen University for assistance in air mass
back trajectory analysis.

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
