# Peer review of "Seasonal variation of aerosol iron solubility in coarse and fine particles at an inland city in northwestern China"

_Atmospheric Chemistry and Physics, 2022_

## Author Comment (AC1)

Comments by referees are in blue.
Our replies are in black.
Changes to the manuscript are highlighted in red both here and in the revised manuscript.

**Reply to referee #1**
 The manuscript investigated total Fe, dissolved Fe and Fe solubility for coarse (>1 μm) and fine (<1 μm) particles in four different seasons at Xi'an, China impacted by anthropogenic emissions and desert dust, combing with the relative humidity and aerosol pH. This work is very useful to realize the importance of RH and aerosol acidity in regulating Fe solubility in atmospheric particles. I would therefore consider the publication of this article once the authors have addressed the following comments.
**Reply:** We would like to thank Ref #1 for recommending our manuscript for final publication after minor revision. His/her comments, which have greatly helped us improve our manuscript, have been adequately addressed in the revised manuscript, as detailed below.
**Major concerns:**
 I think in introduction there are wrong mentions. For example, the organic complexations in L. 74 are not analyzed in following results and discussions. The seasonal variation of Fe solubility, on another hand, has only been explored by a few previous studies as shown in the introduction. Again, you investigated seasonal variations of total Fe, dissolved Fe and Fe solubility in this article. So is it a repetitive work? Please go through the words logically.
**Reply:** In fact we discussed the correlation between Fe solubility and aerosol oxalate in the original manuscript (page 23-24). As a result, it is necessary to mention the effects of organic complexation on aerosol Fe solubility in the introduction.
 As only a few studies examined seasonal variations of aerosol Fe solubility, our work which examines aerosol Fe solubility in Xi'an at different seasons is still novel, considering the importance of aerosol Fe solubility. In addition, compared to those previous studies, our work has further examined mechanisms which drive the observed seasonal variation. In the revised manuscript (page 5-6) we have added the following sentences to further explain what seasonal variation of aerosol Fe solubility can tell us: "For example, desert dust aerosol mainly occurs in spring at Xi'an where our present work was conducted, while anthropogenic emission become more important in winter (Cao and Cui, 2021); furthermore, higher temperature in summer causes more ammonium to partition in the gas phase and thus leads to higher aerosol acidity (Ding et al., 2019; Zhou et al., 2022)."
 Combining with the Fig. 4, 8 and 9, as well as table A1, the aerosol Fe solubility in this field observation is lower than some previous studies based on field data on regions influenced by anthropogenic emission and pyrogenic iron source. Please give the reasons for the relative low Fe solubility in the article. If possible, the authors should cite more literatures and field data at sampling sites, summarizing in forms of table to show total Fe and Fe solubility in next revised manuscript.
**Reply:** As suggested by the referee, in the revised manuscript (page 21-22) we have included a new section (Section 4.1.2, and it two paragraphs) to compare our work with previous studies carried out in northern China.
 It is true that our reported Fe solubility was much lower than those reported by some previous studied. As explained in Section 4.1.2 in the revised manuscript, it is because previous studies used sonication, larger filter pore size and/or stronger leaching solutions, leading to higher Fe solubility. For more details, please kindly refer to our revised manuscript (page 21-22).

**Minor concerns:**

49: "primary production" to "primary productions".

**Reply:** It has been corrected in the revised manuscript (page 4).

50: The authors need to clarify the chemical mechanisms as dissolved iron contributing to ROS formations in aerosols. It is not enough only listing the references. I think Fenton reaction is a good standpoint.

**Reply:** As suggested, in the revised manuscript (page 4) we have made the following change to clarify the mechanism for ROS formation: "…recognized as an important source of reactive oxygen species in aerosol particles via mechanisms such as the Fenton reaction…"

59: "a number of studies have been conducted in the last 2-3 decades." But authors only cite some references in recent years (2018 to 2021) and should replenish more studies in former years than 2010.

**Reply:** As suggested, in the revised manuscript we have cited a few important papers published before 2010.

66: "contribution" to "contributions".

**Reply:** It has been corrected in the revised manuscript (page 5).

74: Irrelevant statement in introduction. From view of this manuscript, the authors aim to study the effect of aerosol acidification on aerosol Fe solubility, rather than organic complexation, as yet it is not shown in entire paper. Please revise it.

**Reply:** In fact we discussed the correlation between Fe solubility and aerosol oxalate in the original manuscript (page 23-24). As a result, it is necessary to mention the effects of organic complexation on aerosol Fe solubility in the introduction.

112: "W41 filter used for aerosol sampling were acid-washed to reduce background levels." I am little confused by the acid-wash and you should explain the pretreatment. Or sampling cut-offs and sampler were acid-washed?

**Reply:** We applied acid-wash to our filters in order to reduce the background. As this was detailed in our previous work (Zhang et al., 2022), our current paper did not describe it in details. For better clarification, in the revised manuscript (page 7) we have changed the sentence to "W41 filters, which were used for aerosol sampling, were acid-washed to reduce background levels."

119, This sentence could be revised as "Each filter was equally halved."

**Reply:** As suggested, it has been corrected in the revised manuscript (page 7).

122, this again confusing. Why did the authors fill Teflon jar with 20 mL HNO3 after acid digestion? Whether the results were same if replacement with ultrapure waters?

**Reply:** After evaporation, we needed to dilute the residual in the Teflon jar to a given volume (20 mL in our work) for further ICP-MS analysis. We used 1% $HNO_3$ instead of ultrapure water because our standards used in ICP-MS analysis also contained 1% $HNO_3$; in addition, using $HNO_3$ (or other strong acids) will prevent re-precipitation of metals in the solutions.

Figure 3 and Figure 7: These plots missed the color legends labeling as the coarse or fine particles.

**Reply:** It is a good suggestion. As the two figures already contains many symbols, in the revised manuscript we have provided additional information in the figure captions: "Blue symbols represent fine particles and brown symbols represent coarse particles." Captions have also been updated for Figures in the supporting information.

369: Misspelling, please revise "Ass shown in Figure S11" to "As shown".

**Reply:** It has been corrected in the revised manuscript (page 26).

405: "Both Shi et al. (2020) and we suggested that high RH could promote Fe dissolution via acid processing." Only relationship between RH and Fe fractional solubility is not persuasively in favour of it. If possible, the writer can add the soluble ion balance (I = 2[SO4] + [NO3]+ [Cl] − 2[Ca] − [NH4] − [Na] − 2[Mg] − [K]), a proxy of the acidification of the aerosol aqueous phase, in Fig. 8 to see if they correlate. When I >0, excess H+ is required in the associated aqueous phase to neutralize the excess anions.

**Reply:** The referee raised an interesting point. Indeed the relationship between RH and Fe solubility was not convincing enough, and this is exactly why we further discussed the dependence of Fe solubility on aerosol pH (calculated using ISORROPIA) in the next two paragraph (line 406-432 in the original manuscript). In response to this comment, in the revised manuscript we have deleted "via acid processing" as the effects of acid processing were specifically discussed in the next two paragraphs.

As it is widely believed that aerosol pH calculated using ISORROPIA, when compared to ion balance, can better represent aerosol acidity, in our work we have chosen to use aerosol pH instead of ion balance.

426: Did the lower aerosol pH exist in summer and autumn? Several studies have shown the formations of secondary inorganic aerosols during heavy haze episodes frequently in winter dominate the higher aerosol acidity in cities, so is conflicting with your findings? Can you give an interpretation for it?

**Reply:** Many studies (e.g., Tao and Murphy, 2019; Yang and Weber, 2022; Zhou et al., 2022) found lower aerosol pH in warm seasons (summer and autumn), as higher temperatures favors $NH_3/NH_4^+$ to partitioning into gas phase and thus leads to higher aerosol acidity (lower aerosol pH). Many studies discussed haze formation during winter when temperature variations are much smaller than those observed for different seasons; during haze formation aerosol sulfate and nitrate were usually increased a lot, leading to higher aerosol acidity. Therefore, lower aerosol pH in summer and autumn (compared to spring and winter) does not conflict with lower aerosol pH during haze events in winter (compared to non-haze periods); in fact, the two aerosol pH variations occurred at different timescales.

Furthermore, we have revised the last paragraph in the revised manuscript (page 31) to further discuss seasonal variation of aerosol acidity, and the referee is kindly referred to our revised manuscript for further information.

429: "the observed higher Fe solubility in summer and autumn" at L. 429 is completely contrary to the wordings at L. 443-444 as "dissolved Fe concentrations were lowest in spring and summer". Please revise this.

**Reply:** The two statements are not contrary in fact, because Fe solubility is not equivalent to dissolved Fe, which also depend on total Fe. Although Fe solubility was high in summer, total Fe was lowest in summer; therefore, dissolved Fe was low in summer. Please refer to Figures 1 and 4 in our original manuscript for more information.

**References:**

Tao, Y. and Murphy, J. G.: The Mechanisms Responsible for the Interactions among Oxalate, pH, and Fe Dissolution in PM2.5, ACS Earth and Space Chem., 3, 2259-2265, 2019.

Yang, Y. and Weber, R. J.: Ultrafiltration to characterize PM2.5 water-soluble iron and its sources in an urban environment, Atmos. Environ., 286, 119246, 2022.

Zhou, M., Zheng, G., Wang, H., Qiao, L., Zhu, S., Huang, D., An, J., Lou, S., Tao, S., Wang, Q., Yan, R., Ma, Y., Chen, C., Cheng, Y., Su, H., and Huang, C.: Long-term trends and drivers of aerosol pH in eastern China, Atmos. Chem. Phys., 22, 13833-13844, 2022.

---

## Author Comment (AC2)

Comments by referees are in blue.
Our replies are in black.
Changes to the manuscript are highlighted in red both here and in the revised manuscript.

**Reply to referee #2**

This study by Zhang et al., investigated the seasonal variation of aerosol Fe solubility for coarse and fine particles at Xi'an. Overall, the manuscript is well organized and the results are clearly presented with comprehensive discussion. The topic of this study is of great interest in the community. The results of this study are interesting and important for the understanding the role of Fe in atmosphere, especially for the biogeochemical cycle of Fe. I am very pleased to recommend this manuscript for publication after a minor revision. My comments are shown as below.

**Reply:** We would like to thank Ref #2 for recommending our manuscript for final publication after minor revision. His/her comments, which have greatly helped us improve our manuscript, have been adequately addressed in the revised manuscript, as detailed below.

Line 140: Gaseous compounds were not considered when calculating aerosol acidity. Clarify if this will affect the trend of aerosol acidity. This is important for the discussion on aerosol acidity. Generally, omission of gaseous compounds will lead systematically underestimation of pH (over estimation of acidity). I believe that the trend of aerosol acidity would not be affected significantly; therefore, the discussion on aerosol acidity is still valid. But a clarification is needed here.

**Reply:** It is true that our calculation may underestimate aerosol pH (as we stated in our original manuscript), and as the referee pointed out, the general trend would not be significantly affected. In the revised manuscript (page 8), we have added one sentence to clarify it: "…this may cause some biases (likely underestimation) in calculated aerosol pH (Guo et al., 2015; Hennigan et al., 2015; Pye et al., 2020), but the overall trend of aerosol pH would not be significantly affected."

Line 146: If the arrival time is 8 am of each day, the number of trajectories should be same as the samples. But the number of trajectories is much more than that of samples. Please verify the accuracy of the description.

**Reply:** In fact we have four back trajectories for each day. In the revised manuscript (page 9) we have revised this sentence to provide accurate description: "Back trajectories were determined with arrival height of 100 m above the ground level and arrival time of 08:00, 14:00, 20:00 and 02:00 on the next day (Wang et al., 2020), and every day four back trajectories were obtained."

It is stated that "…suggesting desert dust always as the dominant source of total aerosol Fe at Xi'an, regardless of particle size range and seasons." (Line 206-207). But in line 202-204, the authors also mentioned anthropogenic emissions as an important factor. The authors need to clarify how they are consistent.

**Reply:** The statement in our original manuscript (line 202-204) was not accurate and appeared to be inconsistent with the statement in line 206-207. In the revised manuscript (page 13), it has been changed to "Overall, compared to summer and autumn, total aerosol Fe were higher in spring and winter when higher aerosol mass concentrations were also observed (Figure S3)."

Line 229-230: Authors indicated that dissolved Fe concentration in winter for Xi'an is higher than that for Qingdao. It may be resulted from the differences of total Fe concentrations in the two cities. Authors can add one sentence here to make an explanation.

**Reply:** The referee is right. As discussed in the original manuscript (Section 3.2), total Fe concentrations were much higher in Xi'an than Qingdao for both coarse and fine particles. In response to this comment, in the revised manuscript (page 15) we have added one sentence to

explain why dissolved Fe concentration in winter for Xi'an is higher than that for Qingdao: "…and one major reason is that total Fe concentrations were significantly higher at Xi'an than Qingdao"

Line 255-268: When investigating the source of soluble Fe, the authors talked about the correlation of soluble Fe with elements like K+, Pb and Al, which is valid. However, the authors started the discussion from K+ without any justifications. The correlation with all elements has been actually listed in Table S3. I suggest the authors to have an overall description of the correlation with all elements before mentioned K+ (with the highest correlation).

Line 261-262: Any literature to support that biomass burning emission is important for autumn and winter in Xi'an?

Line 264: "anthropogenic emission" is a vague description. Please specify it or list some possible anthropogenic sources.

**Reply:** The three comments list above are all related to sources of dissolved aerosol Fe, and thus are replied here together.

As suggested by ref#2, in the revised manuscript (page 16-17) we have discussed correlations between dissolved Fe and several aerosol species (including but not limited to $K^+$) to further use the information provided in Table S3. After revision, we have gained further insights into important sources for dissolved aerosol Fe, including secondary formation, biomass burning, as well as vehicle emission, coal combustion, steel industry and metal smelting. Please refer to the last two paragraphs (page 16-17) in Section 3.3 for more details.

Section 4.1: Authors may need to compare the Fe solubilities in winter between Qingdao and Xi'an, as the comparisons for total and dissolved Fe between these two cities.

**Reply: I**t is a good point. In Section 4.1.2 of the revised manuscript(page 21-22), we have compared wintertime results between Qingdao and Xi'an, and briefly discussed the possible reasons for the difference.

Line 308-310: This sentence does not provide sufficient information. To my understanding, the reverse relationship actually reflects the different source (or affecting factor) of total Fe and water-soluble Fe. The authors can try to explain the mechanism or just state that the mechanism needs to be further investigated.

**Reply:** In fact several mechanisms could possibly explain such inverse relationship, as discussed in a few review papers (Mahowald et al., 2018; Meskhidze et al., 2019). It may not be necessary in our manuscript to repeat these explanations. Instead, in the revised manuscript (page 19) we have added two references from which interested authors can find more information: "Several mechanisms can qualitatively explain such inverse dependence, but a consensus has not been reached yet (Mahowald et al., 2018; Meskhidze et al., 2019)."

Line 383: I am not sure if the description of "Secondary formation of dissolved aerosol Fe" is accurate. I understand that secondary process may promote the dissolve of Fe but this description may be misleading.

**Reply:** In our work we use primary sources to represent dissolved Fe associated with un-processed mineral dust and anthropogenic particles, and secondary source to represent dissolved Fe which is formed via dissolution of insoluble Fe by chemical aging. To increase clarity, in the revised manuscript (page 28) we have made the following modification: "…indicate the importance of secondary formation of dissolved aerosol Fe (i.e. dissolution of insoluble Fe to dissolved Fe via aging processes)."

Line 409-410: Authors stated that Fe solubility continuously decreased with increasing aerosol pH (from <2 to >5) for fine particles, this trend is generally right. But if we see Figure 9,

Fe solubility slightly increased with pH from <3 to >3 for fine particles, so please make your description be more conservative.

**Reply:** The referee is right. The word "continuously" is not accurate. In order to make our statement more accurate, in the revised manuscript (page 30) we have changed this sentence to "For fine particles, Fe solubility in general decreased with increasing aerosol pH (from <2 to >5)".

Authors indicated that desert dust was not the main source of dissolved Fe, and chemical aging showed a small impact on Fe solubility for fine particles. It may imply that anthropogenic emission is the dominant source of dissolved Fe in fine particles. Authors can add some sentences in section 6 to illustrate it and make your conclusions be more specific.

**Reply:** We would like to thank ref #2 for his/her kind suggestion. We have thought about making this conclusion more specific and stronger. However, as we do not have direct or quantitative results, we would like to be conservative at this point and thus no change has been made.

Some other minor issues as listed below:

1) The title of section 3.1 may be more suitable with "Meteorological conditions and particulate matter concentrations" since particulate matter concentrations are also discussed in this section.

**Reply:** As suggested, the title has been changed to "Meteorological conditions and aerosol concentrations" in the revised manuscript ( page 9).

2) Line 109: Please delete the second word of "and" in "Coarse (>1 μm) and fine (<1 μm) and aerosol particles".

**Reply:** The redundant "and" has been deleted in the revised manuscript.

3) Line 110: Specify the time is am or pm.

**Reply:** The time has been specified in the revised manuscript (page 7): "…from 08:00 am to 07:30 am next day…"

4) Line 124: Specify the model of the ICP-MS and the MDL.

**Reply:** In the revised manuscript (page 8) more information has been provided for the ICP-MS we used: "…using inductively coupled plasma mass spectrometry (iCAP Q, Thermo Fisher Scientific, USA)."

5) Line 144: Verify the time resolution is 3 or 6 hours.

**Reply:** The time resolution is actually 3 h, and in the revised manuscript (page 9) we have corrected it.

6) Line 185 and line 229: Please replace the Chinese character in parentheses with "and".

**Reply:** These two errors have been corrected in the revised manuscript (page 11 and page 15).

7) Line 213: Correct the typo "deust".

**Reply:** This typo has been corrected in the revised manuscript (page 13).

8) Line 356: Space should be added between the parentheses and word.

**Reply:** The suggested corrections have been implemented in the revised manuscript.

9) Line 405: Change "we" to "the current study".

**Reply:** As suggested, in the revised manuscript (page 29) this sentence has been changed to: "Therefore, both Shi et al. (2020) and our present work suggested that…"

10) Line 425: The values of Fe solubility for coarse particles in parentheses were wrong. Please follow Table A1 and revise them.

**Reply:** We have carefully checked all the numbers in parentheses in this paragraph, and made necessary corrections in the revised manuscript (page 31).

---

## Author Comment (AC3)

Comments by referees are in blue.
Our replies are in black.
Changes to the manuscript are highlighted in red both here and in the revised manuscript.

**Reply to referee #3**
**General comments**

The study reports seasonal variation of total and dissolved Fe concentrations and fractional solubility in coarse and fine particles collected in Xi'an. More than 120 samples collected over a year are a valuable data set on Fe solubility in East Asia, which can contribute to a better understanding of the factors controlling Fe solubility. However, these valuable data sets were not fully utilized in the paper, as the discussion in this paper is mainly about averages (or medians) for concentrations and Fe solubility. There was no discussion of total and dissolved Fe or its solubility during the two Asian dust events, which have the potential to be important Fe supply events to the ocean surface. Furthermore, Fe solubility in some of the winter samples exceeded 5%, and the reasons for their high solubility should be mentioned. It would be desirable to discuss the daily variation of Fe concentration and Fe solubility in each season to solve these problems.

**Reply:** In addition to average and median values, in our original manuscript we also discussed the dependence of Fe solubility on air mass sources in different seasons (Section 4.2), and the dependence of Fe solubility on chemical aging in different seasons (Section 4.3).

It is a good idea to discuss Fe solubility during dust events and winter haze periods. In response to this comment, in the revised manuscript (page 22-23) we have added a new section (Section 4.1.3, and it has two paragraphs) to discuss Fe solubility during dust events in spring and high-RH haze periods in winter. Please kindly refer to our revised manuscript for more details.

The importance of anthropogenic Fe on enhancement of Fe solubility in fine aerosol particles were reported by this study. However, the discussions on the emission sources of anthropogenic Fe and its tracer elements are a little bit broad. For instance, the dominant source of anthropogenic Fe in coarse and fine aerosol particles was biomass burning due to a good correlation between dissolved Fe and K+ concentrations. It is suspicious that K+ in coarse particles is from biomass burning because aerosols from high-temperature combustion, including biomass burning, are abundant in fine aerosol particles. As the authors said in the introduction, emission sources and physicochemical properties are different between coarse and fine aerosol particles. Therefore, discussions on the emission source of anthropogenic Fe and its tracer elements in each size fraction is required. I think that Table S3 is supported to evaluate the emission source of anthropogenic Fe.

**Reply:** In our original manuscript, indeed our discussion related to sources of dissolve Fe was a little bit broad.

As suggested by ref#3, in the revised manuscript (page 16-17) we have discussed correlations between dissolved Fe and several aerosol species (including but not limited to $K^+$) to further use the information provided in Table S3. After revision, we have gained further insights into important sources for dissolved aerosol Fe, including secondary formation, biomass burning, as well as vehicle emission, coal combustion, steel industry and metal smelting. Please refer to the last two paragraphs (page 16-17) in Section 3.3 for more details.

The relationship between aerosol pH and iron solubility is the most important topic in this paper. Aerosol pH in coarse aerosol particles is usually 1 to 4 units higher than that in fine aerosol particles because non-volatile cation (e.g., Na, K, Mg, and Ca) is mainly present in coarse aerosol particles. However, the median aerosol pH in coarse aerosol particles was almost the same or lower than those in fine aerosol particles. If the low pH of the coarse particles is due to a bias in the

thermodynamic calculations, it may overestimate the importance of acidic processes in the coarse particles. Please explain the reason for lower pH in coarse aerosol particles than in fine aerosol particles.

**Reply:** We agree with the referee 1) that aerosol pH are generally expected to be higher for coarse particles than fine particles, and 2) that similar or lower aerosol pH were reported by our work for coarse particles (when compared to fine particles) in spring and autumn. The reasons are not clear yet (likely due to uncertainties in aerosol pH calculation for coarse particles), though we note that lower aerosol pH values were also reported in previous work for coarser particles. The first step is to reduce uncertainties in aerosol pH calculation via concurrent measurement of gaseous species (e.g., $NH_3$ and $HNO_3$).

In the revised manuscript (page 8-9) we have added a few sentences to discuss these caveats: "The effects of $NH_3(g)$ and $HNO_3(g)$ were not taken into account as their concentrations were not measured; this may cause some biases (likely underestimation) in calculated aerosol pH (Guo et al., 2015; Hennigan et al., 2015; Pye et al., 2020), but the overall trend of aerosol pH would not be significantly affected. The reverse mode was not used in our work, as results calculated using the reverse mode are very sensitive to uncertainties in concentrations of common aerosol ions (Hennigan et al., 2015). Coarse particles are generally expected to be less acidic than fine particles, and it is not clear yet why similar and even lower aerosol pH were observed for coarse particles (when compared to fine particles) in spring and autumn at Xi'an (Figure S14). This may be caused by biases in aerosol pH calculation, and lower aerosol pH values were also reported in previous work for coarser particles carried out at northern Colorado, United States (Young et al., 2013). Concurrent measurements of gaseous NH3, HNO3 and HCl would help reduce uncertainties in calculated aerosol pH and have been implemented in our following studies since July 2022.."

I am also concerned that the Fe solubility reported by this study was much lower than those in aerosol particles collected in China by previous studies. High Fe solubility (>10%) has been observed in aerosol particles collected in China, including Xi'an, but such high Fe solubilities were not found in this study. It is necessary to explain why Fe solubility differs significantly from previous studies, even though the aerosols were collected in the same city.

**Reply:** As suggested by the referee, in the revised manuscript (page 21-22) we have included a new section (Section 4.1.2, and it two paragraphs) to compare our work with previous studies carried out in northern China.

It is true that our reported Fe solubility was much lower than those reported by some previous studied. As explained in Section 4.1.3 in the revised manuscript, it is because previous studies used sonication, larger filter pore size and/or stronger leaching solutions, leading to higher Fe solubility. For more details, please kindly refer to our manuscript (page 21-22).

Since there are several concerns, I cannot recommend the publication of this study in ACP in its current state. However, this manuscript would represent a valuable contribution to the field of aerosol research after major revisions because there are few studies on seasonal variation of Fe solubility and its controlling factors.

**Reply:** We highly appreciate his/her very positive evaluation of our manuscript as well as thorough and insightful comments. We can see that the referee has spent a lot of time on reviewing our manuscript. We have considered all the comments very seriously and addressed them carefully, and revised our manuscript accordingly. These comments have helped us significantly improve our manuscript. If the referee has further comments, we will be very glad to address them.

**Specific comments**

L81–83: Please provide specific examples of how seasonal variations of emission sources and chemical processes of Fe affect its solubility.

**Reply:** That is a very good idea. In the revised manuscript (page 5) we have added the following sentences to provide a few examples: "For example, desert dust aerosol mainly occurs in spring at Xi'an where our present work was conducted, while anthropogenic emission become more important in winter (Cao and Cui, 2021); furthermore, higher temperature in summer causes more ammonium to partition in the gas phase and thus leads to higher aerosol acidity (Ding et al., 2019; Zhou et al., 2022)."

L95: Many readers are not familiar with Xi'an and its surrounding topography. Please provide a map showing the sampling site and the surrounding area.

**Reply:** This is a good suggestion. In the revised manuscript we have provided two maps (Figure S1) in the supporting information, and also revised the main text (page 6) accordingly: "Xi'an has a population of ~13 million and is located in the middle of the Guanzhong Plain which is surrounded by Qinling Mountains and Chinese Loess Plateau (Figure S1)…"

L121: Please clarify acid compositions.

**Reply:** As digestion was detailed in our previous study (Zhang et al., 2022), we only describe the experimental procedure briefly in the present paper and kindly refer readers to our previous study (Zhang et al., 2022) for more information.

L124–126: How were recoveries of target elements determined? If you measure certificated reference material, please specify it.

Reply: Indeed certificated reference materials were used, and in the revised manuscript (page 8) we have provided relevant information: "…the recovery rates were found to be 90-110% for Fe using certificated reference materials (GBW07454 and GSB07-3272-2015)"

L135–136: If gas species is not employed as an input parameter, the reverse mode would be the appropriate calculation method.

**Reply:** We did not use the reverse mode, as results calculated using the reverse mode are very sensitive to uncertainties in concentrations of common aerosol ions (Hennigan et al., 2015). In the revised manuscript (page 8-9) we have added one sentence to explain explicitly why the reverse mode was not used in our work: "The reverse mode was not used in our work, as results calculated using the reverse mode are very sensitive to uncertainties in concentrations of common aerosol ions (Hennigan et al., 2015)."

In order to reduce the uncertainties in calculated aerosol pH, measurements of gaseous species are necessary and have been implemented in our later studies since July 2022, as we pointed out in the revised manuscript (page 9): "Concurrent measurements of gaseous $NH_3$, $HNO_3$ and HCl would help reduce uncertainties in calculated aerosol pH and have been implemented in our following studies since July 2022."

L161–163: I guess that the standard deviation of aerosol mass concentration in spring is too small for the concentration range of PM10. Please confirm it.

**Reply:** The referee is right, and the standard deviation should be 61. We have corrected it in the revised manuscript (page 10).

L182–186: Why were the Fe concentrations of aerosol particles collected in Xi'an higher than those collected in Qingdao? Is Qingdao classified as an industrial, residential, or background area?

**Reply:** Qingdao is a coastal city in the northern China. The main reason for higher total Fe concentrations in Xi'an, when compared to Qingdao, is that aerosol mass concentrations are much higher in Xi'an. In the revised manuscript (page 15) we have added one sentence to explain in

brief why total Fe concentrations were higher in Xi'an: "…mainly because during wintertime aerosol mass concentrations were much higher at Xi'an than Qingdao."

L:185: There is a garbled text between 1831±866 and 2058±1037 ng/m3. The same garbled text was found in line 229.

**Reply:** These two errors have been corrected in the revised manuscript (page 11 and page 15).

L190–195: Iron in mineral dust is mainly present in coarse aerosol particles. Indeed, the contribution of aerosol mass concentration of coarse aerosol particles in spring and Asian dust events was larger than in other seasons due to the low PM2.5/PM10 ratio. However, the contribution of Fe in coarse aerosol particles to total Fe in TSP was lower than in other seasons. This result is inconsistent with a mass ratio of PM2.5/PM10. Please explain the reason for the lowest contribution of Fe in coarse aerosol particles to that in spring samples. In addition, nss-Ca, Al, and Ti are good tracers of Asian dust. Are these elements in the aerosol collected in the spring also present mainly in the fine particles?

**Reply:** We were also quite surprised to notice that the average contribution of coarse particles to total Fe in TSP was lowest in spring when the influence of desert dust aerosol was largest. As suggested by the referee, we further found that the average contribution of coarse particles to total Al in TSP was also lowest in spring. In addition, during dust periods the average contributions of coarse particles to Fe and Al in TSP were even slightly smaller than the average values in spring. We tentatively believe that is because for desert dust aerosol, Fe and Al were enriched in fine particles while the major component in coarse particles was quartz; of course, this is subject to further discussion and investigation.

Accordingly, in the revised manuscript (page 12) we have made the following change: "The average contribution of coarse particles to total Fe in TSP (total suspended particles) were 29%, 57%, 63% and 47% in spring, summer, autumn and winter, being lowest in spring when the influence of desert dust aerosol was largest. Statistical analysis (paired t-test) suggested that compared to fine particles, total Fe in coarse particles was significantly lower in spring (p<0.01, α=0.05) while significantly higher in summer and autumn (p<0.01, α=0.05); in addition, there was no significant difference between coarse and fine particles in winter (p=0.13, α=0.05). The average contribution of coarse particles to total Al in TSP were 26%, 50%, 54% and 40% in spring, summer, autumn and winter, also being lowest in spring. Furthermore, during dust periods (12-17 and 27-30 April) the average contribution of coarse particles was found to be 24% for total Fe and 23% for total Al in TSP, even slightly smaller than the average values in spring. This is probably because for desert dust aerosol, Fe and Al were enriched in fine particles while the major component in coarse particles was quartz (Journet et al., 2014); nevertheless, further measurements are needed to better understand size distribution of trace metals in desert dust aerosol."

L199: Aerosol particles collected in North America and the Atlantic Ocean are unlikely appropriate comparison targets of Fe solubility in aerosol particles collected in China. I would suggest a comparison with dissolved iron in aerosols collected in Xi'an (e.g., He et al., 2021; Lei et al., 2023).

**Reply:** Here in our original manuscript we compared our observed seasonal variation of total Fe at Xi'an with those reported by Yang et al. (2020) on an Island in China and by Chen and Siefert (2004) over the Atlantic Ocean. Seasonal variations obtained at different regions can be comparable, though it may not be appropriate to compare actual concentrations or solubility, as pointed out by the referee.

We also would like to thank the referee for bringing our attention to the two papers (He et al., 2021; Lei et al., 2023). We have added a new section (Section 4.1.2) in the revised manuscript

(page 21-22) to compare Fe solubility reported in our present work with those reported in previous studies carried out in China (including the two papers mentioned by the referee). As the referee raised similar concerns in a previous comment, we kindly refer him/her to the revised manuscript and our response to the previous comment for more details.

L208: In general, [Fe]/[Al] means molar concentration ratio rather than mass concentration ratio. Please removed the bracket.

**Reply:** We have removed all the brackets in the revised manuscript. Furthermore, as suggested by the referee, in the revised manuscript and SI we always A/B to represent mass ratios and [A]/[B] to represent molar mass ratios.

L258–261: Coal combustion is also a dominant source of K+ in fine aerosol particles. Indeed, dissolved Fe concentration correlated with Pb as a tracer element of coal combustions. In addition, fuel oil (e.g., heavy oil, gasoline, and diesel) combustions emit dissolved Fe. Furthermore, the steel industry is the dominant source of anthropogenic Fe-oxides. A more detailed discussion of the emission sources of dissolved Fe in aerosol particles is needed.

L262–265: Please discuss in detail regarding emission sources of Pb and dissolved Fe in coarse and fine aerosol particles. Lead in fine aerosol particles is derived from high-temperature combustions (e.g., coal combustions and municipal solid waste incineration). In contrast, Pb in coarse aerosol particles is derived from the resuspension of road dust (e.g., the pigment of road paint). Therefore, emission sources of dissolved Fe are likely different between coarse and fine aerosol particles.

**Reply:** The two comments are both directly related to sources of dissolved aerosol Fe, and are thus addressed together.

As suggested by ref#3, in the revised manuscript (page 16-17) we have discussed correlations between dissolved Fe and several aerosol species (including but not limited to $K^+$) to further use the information provided in Table S3. After revision, we have gained further insights into important sources for dissolved aerosol Fe, including secondary formation, biomass burning, as well as vehicle emission, coal combustion, steel industry and metal smelting. Please refer to the last two paragraphs (page 16-17) in Section 3.3 for more details.

We carried out literature survey and also consulted a few colleagues who knows aerosol metal sources very well. We are not sure if coal combustion is an important source of aerosol $K^+$. If ref #3 could provide some literature, we can further revise our manuscript. We have also revised manuscript to discuss the correlation between Pb and dissolved Fe, and please kindly refer to our revised manuscript (page 17) for more details.

L305–306: Please explain why the inverse relationship between Fe solubility and total Fe in autumn samples could not be observed. In addition, it seems that the constant of a and the power of b in equation 1 has seasonal dependence. Please discuss the reason for the seasonal variation of the fitting equation.

**Reply:** First of all, several mechanisms could possibly explain such inverse relationship, but we have not reached a consensus. In the revised manuscript (page 19) we have added a sentence for further explanation of such inverse dependence: "Several mechanisms can qualitatively explain such inverse dependence, but a consensus has not been reached yet (Mahowald et al., 2018; Meskhidze et al., 2019)."

Our main purpose of line 305-306 in the original manuscript is to state clearly (and remind interested colleagues) that such inverse dependence is not universe, though widely reported. However, it is not clear yet why such inverse dependence was not observed in some studies. In response to the referee, in the revised manuscript (page 20) we have added one sentence to

acknowledge that it is not clear yet why such inverse dependence was not observed in some studies: "It is not clear yet why such inverse dependence was not observed in these studies."

We would like to thank the referee for bringing our attentions to seasonal variations of the two constants. In addition to seasonal variations, we further find that these values also differ between fine and coarse particles. We have carefully thought about the physical meanings of these two constants: the physical meaning of *a* is not obvious, while *b* represents the sensitivity of Fe solubility to relative change in total Fe concentration. In the revised manuscript (page 19) we have added the following sentence to explain the physical meaning of *b*: "…where $f_s$(Fe) is Fe solubility and $[Fe]_T$ is total Fe concentration, and *b* represents the sensitivity of Fe solubility to relative change in total Fe concentration."

Furthermore, in the revised manuscript (page 21) we have added one paragraph to discuss seasonal and size variations of *b* values: "A larger b value means that Fe solubility is more sensitive to relative change in total Fe concentration. For our measurements conducted at Xi'an, b values were determined to be 0.30, 0.23 and 0.91 in spring, summer and winter for fine particles (Figure S7), and 0.52, 0.65 and 1.16 for coarse particles (Figure 5). One can see that the b values in winter were much larger than those in spring and summer for both fine and coarse particles; furthermore, in each of the three seasons (spring, summer and winter), the b value was larger for coarse particles than fine particles." Nevertheless, currently we do not how to explain seasonal and size dependence of *b* values.

L347–349: Why not evaluate the relationship of Fe solubility with nitrate and sulfate separately? Identifying the acid species that increase iron solubility is one of the important topics. **Reply:** We indeed evaluated the relationship of Fe solubility with nitrate and sulfate separately, though these results are not shown in our original manuscript. In the revised SI, we have included two figures to show the correlations of Fe solubility with [sulfate]/[Fe] (Figure S10) and [nitrate]/[Fe] (Figure S11). In addition, in the revised manuscript (page 26) we have added one sentence to describe the overall result which we can obtain from these two figures: "In addition, as shown in Figures S10-S11, correlations of Fe solubility with [nitrate]/[Fe] were better than (or very similar to) these with [sulfate]/[Fe]for coarse particles in the four seasons, whereas no obvious trend was not observed for fine particles."

L369: Ass rephrased by as. **Reply:** This typo has been corrected in the revised manuscript (page 27).

L414–416: Please explain why Fe solubility was low (<1%) in some aerosol particles despite low pH and high RH. Also, were the aerosol samples with lower Fe solubility at low pH and high RH found in a particular season or in any season? **Reply:** These samples were found in all the four seasons, though their occurrence in winter was less frequent. It is not clear why low Fe solubility still occurred despite high RH and aerosol acidity, but Fe mineralogy may be a key reason. In the revised manuscript (page 30) we have added a few sentences for further discussion: "In total 34 samples for coarse particles (9 in spring, 8 in summer, 12 in autumn and 5 in winter) and 18 samples for fine particles (7 in spring, 6 in summer, 4 in autumn and 1 in winter) fulfilled the above conditions (pH<4, RH>80%, and Fe solubility <1%). Fe mineralogy may possibly explain the observed low Fe solubility despite high RH and aerosol acidity, and concurrent measurements of Fe mineralogy could provide further clues."

L416-419: Specific surface area is one of the factors controlling fractional Fe solubility in aerosol particles (Baker and Jickells, 2006; McDaniel et al., 2019). The specific surface area of fine aerosol particles is usually higher than that of coarse aerosol particles, indicating that fine aerosol particles are more reactive than coarse aerosol particles. Therefore, the fact that the Fe

solubility of fine aerosol particles is higher than that of coarse aerosol particles does not guarantee that the contribution of anthropogenic Fe to Fe solubility is greater.

**Reply:** We agree with the referee. In response to this comment, in the revised manuscript (page 30) we have changed this sentence to make our statement more conservative and to mention the paper by McDaniel et al. (2019): "If we assume at the same pH range Fe solubility enhancement by acid processing was similar for fine and coarse particles, the results displayed in Figure 9 may imply that anthropogenic and pyrogenic Fe played a more important role in Fe solubility enhancement in fine particles at Xi'an, when compared to coarse particles. Mcdaniel et al. (2019) found that soluble Fe concentration was strongly correlated with aerosol surface area for size-resolved aerosol samples collected from several different regions, and suggested surface area as the main factor which affected Fe solubility; they further suggested that this was because Fe solubility enhancement by acid processing could be more effective for aerosol particles with larger surface area and thus smaller particle size."

We were aware of the work by Mcdaniel et al. (2019) quite a while ago, and to be honest we do not quite agree with them; nevertheless, we fully realize that the mechanism we proposed in our current manuscript also has a caveat. For aerosol Fe solubility which is important but quite uncertain, it can be very useful for advancement of our understanding to propose different mechanisms which will be examined by further work.

The seminal work by (Baker and Jickells, 2006) has been cited in the introduction, and is thus not cited here. One reason is that their explanation of the observed size dependence of Fe solubility may not be correct, as discussed in several previous papers (for example, Mahowald et al., 2018; Shi et al., 2011b).

L425–429: Indeed, aerosol pH in spring was higher than those in autumn, but median temperature and RH were almost the same between spring and autumn. Therefore, temperature and RH are not the reason for the higher pH of spring than autumn. One possible reason for high aerosol pH in spring is the large abundance of CaCO3 in Asian dust that acts as buffer species. Please discuss the seasonal variability of aerosol pH with accurate descriptions of the relationship of aerosol pH with temperature, RH, non-volatile cation concentration, etc.

**Reply:** The referee's comment on seasonal variation of aerosol pH is absolutely correct, and desert dust had large effects on aerosol pH in spring at Xi'an. In our original manuscript we did not discuss explicitly seasonal variation of aerosol pH as this is not the focus of manuscript, though it seems that we imply temperature as the main reason for seasonal variation of aerosol pH.

In response to this comment, we have added a few sentence in the revised manuscript (page 31) to discuss briefly while explicitly seasonal variations of aerosol pH at Xi'an: "Compared to summer and autumn, lower temperature in winter favored partitioning of ammonium in aerosol particles and thus led to higher aerosol pH. Average temperatures were similar in spring and autumn at Xi'an (Table S1), but aerosol pH was higher in spring than autumn (Table S4). Higher aerosol pH in spring at Xi'an, when compared to autumn, was caused by increase of non-volatile cations in spring due to the occurrence of desert dust aerosol; in fact, we found that abundance of $Ca^{2+}$ (relative to sulfate) was much higher in spring for both fine and coarse particles."

L427: Please rephrase Table S5 by Table S4.

**Reply:** We have corrected it in the revised manuscript (page 31).

Figure 1: Does the solid line represent the median and the closed square the mean? Please provide legends for the median and average (as well as other box plots).

**Reply:** Some box plots in our original manuscript (Figures 6, 8 and 9) have already had legends. In the revised manuscript all the box plots have been updated to include legends for figures in the main manuscript and SI.

Figure 3: Several plots in panels (c) and (d) are shown with cross symbol. What does the cross symbol represent? Cross symbols can be found in Figures 7, S6 and S11.

**Reply:** Cross symbols represent data points which are not included in fitting. For the revised manuscript, we have include the following sentence in all the relevant figure captions to make this clear: "Cross symbols represent data points which are not included in fittings."

Table S3. More detailed discussion on correlation of dissolved Fe concentration with trace elements are required to estimate anthropogenic Fe source. Are correlation coefficients higher than 0.5 listed in bold? If so, please specify that. Also, please bold the correlation coefficient for Pb in fine grains collected in winter.

**Reply:** As suggested, we have made substantial change to Section 3.3 in the revised manuscript (page 17) in order to further discuss these correlations (between dissolved Fe and trace elements) and sources of dissolved Fe. As the referee also had similar comments previously, he/she is kindly referred to our reply to these comments for further information.

Indeed we highlight all the coefficients higher than 0.5 in bold. In the revised SI we have updated the table caption to provide such information: "Correlation coefficients (R) for dissolved Fe with other species in different seasons. In this table, R values which are >0.5 are highlighted in bold." In addition, the R value for Pb in fine particles collected in winter are highlighted in bold in the revised SI.

Figure S7. This figure is not cited in the main text.

**Reply:** We have included new figures in the supporting information, and updated the main text accordingly. All the figures and tables in the supporting information have been mentioned in the revised manuscript.